# The language of prompting:
# What linguistic properties make a prompt successful?

**Alina Leidinger[*], Robert van Rooij and Ekaterina Shutova**
Institute for Logic, Language and Computation, University of Amsterdam

## Abstract

The latest generation of LLMs can be prompted to achieve impressive zero-shot or few-shot performance in many NLP tasks. However, since performance is highly sensitive to the choice of prompts, considerable effort has been devoted to crowd-sourcing prompts or designing methods for prompt optimisation. Yet, we still lack a systematic understanding of how linguistic properties of prompts correlate with task performance. In this work, we investigate how LLMs of different sizes, pre-trained and instruction-tuned, perform on prompts that are semantically equivalent, but vary in linguistic structure. We investigate both grammatical properties such as mood, tense, aspect and modality, as well as lexico-semantic variation through the use of synonyms. Our findings contradict the common assumption that LLMs achieve optimal performance on lower perplexity prompts that reflect language use in pretraining or instruction-tuning data. Prompts transfer poorly between datasets or models, and performance cannot generally be explained by perplexity, word frequency, ambiguity or prompt length. Based on our results, we put forward a proposal for a more robust and comprehensive evaluation standard for prompting research[1].

## 1 Introduction

NLP has witnessed a rapid succession of large pre-trained language models (LLMs) being released accompanied by reports of impressive performance on a multitude of tasks (Brown et al., 2020; Touvron et al., 2023; Zhang et al., 2022, i.a.). Many works show that increasing model scale decreases pre-training loss and improves downstream task performance *on average* (Brown et al., 2020; Rae et al., 2022). However, this does not hold in general across all samples and instructions (Ganguli et al.,

2022; Sanh et al., 2022). Mounting evidence of performance variability (Köksal et al., 2022; Gonen et al., 2022, i.a.) has been met with an abundance of proposed methods for automatic prompt[2] generation (Shin et al., 2020; Gao et al., 2021; Liu et al., 2023, i.a.) paired with an ongoing discussion on the superiority (or inferiority) of expert-written prompts over generated ones (Logan IV et al., 2022; Webson and Pavlick, 2022). Fundamentally, such variability in performance across prompting strategies raises the question of how LLMs process prompts based on language seen during training. Do they conform to linguistic intuition and respond better to lower perplexity prompts featuring straight-forward sentence structures, frequent and less ambiguous words, or language that has been seen during instruction-tuning?

To test this hypothesis, we examine prompting performance variability in LLMs through the lens of linguistics. To the best of our knowledge, we conduct the first controlled study of LLM performance variability across semantically equivalent prompts that differ in linguistic structure. Specifically, we manually construct parallel sets of prompts that vary systematically in grammatical mood, tense, aspect and modality (550 prompts in total[3]). We study the influence of word frequency and ambiguity by exchanging content words for alternative synonyms. We evaluate five LLMs of different sizes, both instruction-tuned and not instruction-tuned: LLaMA 30b (Touvron et al., 2023), OPT 1.3b, 30b (Zhang et al., 2022) and OPT-IML 1.3b and 30b (Iyer et al., 2022). We focus on understanding of instructions, and therefore, evaluate our models in a zero-shot fashion. We experiment with six datasets for three different tasks: sentiment classification, question answering and natural language inference (NLI). For the instruction-tuned models, our choice of tasks cov-

---

[*] Corresponding author: a.j.leidinger@uva.nl.

[1] All resources are available at: https://github.com/aleidinger/language_of_prompting/

---

[2] We use the terms *prompt* and *instruction* interchangeably.
[3] We release all prompts for all tasks in Appendix D.

ers the fully supervised, cross-dataset and cross-task setting.

Overall, we observe large performance variation due to changes in linguistic structure of the prompt (§5); and this holds even for instruction-tuned models on seen tasks. Furthermore, contrary to previous findings (Gonen et al., 2022), model performance does not appear to correlate with perplexity of the prompts (§6). Further, we find no correlation between performance and prompt length, word sense ambiguity or word frequency. In many cases more complex sentence structures and rare synonyms outperform simpler formulations. Our findings contradict the universal assumptions that LLMs perform best given low perplexity prompts featuring words which we assume (or know) to be frequent in pre-training and instruction-tuning data. This stresses the need for further research into the link between statistical distribution of language at different training stages and model behaviour.

With regards to evaluation practices in the field, our work highlights the limitations of benchmarking multiple models on single, fixed prompts and reporting best-case performance. Prompts generally transfer poorly between datasets even for the same model, let alone across models (§5.2). Instruction-tuning (§5.3) or increasing model size (§5.4) do not preclude the possibility of considerable performance variation, even on seen tasks. These findings, coupled with the fact that many works do not release their prompts, make the results of existing evaluations less reliable and difficult to reproduce. In Section 7, we put forward a proposal for a more robust and comprehensive evaluation framework for prompting research.

## 2 Related work

**Instability in prompting** Papers accompanying newly released models, rarely report prompts used for their evaluation and, typically, do not evaluate on multiple prompts (Brown et al., 2020; Chowdhery et al., 2022; Rae et al., 2022). Sanh et al. (2022) stand alone in reporting performance variation across prompts at the release of T0.

To date, few works have investigated robustness to different prompt formulations. Ishibashi et al. (2023) show that machine-generated prompts are not robust to token deletion or reordering. Webson and Pavlick (2022) evaluate pre-trained and instruction-tuned models on NLI in the few-shot setting. They find that while instruction-

tuning helps robustness against prompt variation, instruction-tuned models respond favourably even to misleading instructions. Shaikh et al. (2023) show that GPT-3 scores drastically worse on bias and toxicity challenge sets with the addition of *'Let's think step by step.'* to a given prompt for chain-of-thought reasoning (Kojima et al., 2022). Razeghi et al. (2022) find that performance on arithmetic tasks correlates with frequency of integers in the training data. Perhaps closest to our work, Gonen et al. (2022) find that lower perplexity of the prompt correlates with higher performance for OPT (Iyer et al., 2022) and BLOOM (Scao et al., 2022) on a variety of different tasks.

In priming or in-context learning, LMs profit from being shown the required input-output format, the distribution of inputs and the label space, while ground truth labels don't seem to be required (Min et al., 2022). Performance is also sensitive to the ordering of demonstration examples (Lu et al., 2022; Zhou et al., 2022; Zhao et al., 2021). Chinchilla performs better on abstract reasoning tasks when test samples cater to prior knowledge acquired through pretraining (Dasgupta et al., 2022).

**Prompting evaluation practices.** Cao et al. (2022) point out that evaluating models on the same prompt does not make for a direct comparison, since models' exposure to different pretraining data results in different responses to individual prompts. Ishibashi et al. (2023) find that machine-generated prompts do not achieve equal performance gains across datasets for the same task. Holtzman et al. (2021) posit that subpar performance of LMs is due to different viable answers outside the answer choices competing for probability mass ("surface form competition"), reducing the score for the correct answer among answer choices. They mediate this using Domain Conditional PMI. Zhao et al. (2021) observe that models overpredict label words that occur more frequently in a prompt, at its end, or are frequent in the pretraining data. They propose fitting an affine function to the LM scores, so that answer options are equally likely for 'content-free' dummy examples.

Contrary to other works (Gonen et al., 2022; Sorensen et al., 2022; Liao et al., 2022) we do not aim at proposing prompt selection methods or calibrate predictions. While many (semi-)automatic approaches to generating prompts have been proposed (Liu et al., 2023; Shin et al., 2020; Jiang et al., 2020; Gao et al., 2021, i.a.), we resort to crafting

prompts manually so as to maintain fine-grained control over sentence structures in our prompts. Logan IV et al. (2022) provide evidence that manually written prompts (Schick and Schütze, 2021) can yield better result than automatically sourced prompts. Further, Ishibashi et al. (2023) point out that automatically generated prompts contain atypical language use, punctuation or spelling mistakes and generalise poorly across datasets. To isolate the effect of individual instructions we restrict ourselves to the zero-shot setting and do not include any demonstration examples in-context. Indeed, LMs have been shown to perform reasonably well in the few-shot setting given unrelated or misleading instructions (Webson and Pavlick, 2022). Similarly, we abstain from prompt-tuning on demonstration examples, so as to not introduce additional sources of variance (Cao et al., 2022).

## 3 Tasks and datasets

To draw robust conclusions across tasks, we conduct experiments on multiple datasets, namely Stanford Sentiment Treebank (SST-2; Socher et al., 2013) and IMDB (Maas et al., 2011) (Sentiment Analysis), SuperGLUE (Wang et al., 2019) Recognizing Textual Entailment (RTE; Wang et al., 2018) and CommitmentBank (CB; De Marneffe et al., 2019) (NLI), Boolean Questions (BoolQ; Clark et al., 2019) and AI2 Reasoning Challenge Easy (ARC-E; Clark et al., 2018) (Question Answering). Our guiding principle in choosing datasets was to have a diverse set of tasks on which pre-trained models such as LLaMA or OPT achieve above-chance performance in the zero-shot setting (Iyer et al., 2022). For OPT-IML, our choice covers the supervised, cross-dataset, and cross-task setting. The OPT-IML models have been instruction-tuned on IMDB and SST using prompts from Prompt-Source (Sanh et al., 2022) and FLAN (Wei et al., 2021). They have been tuned on QA datasets such as SciQ, but not on BoolQ or ARC-E (Iyer et al., 2022) which we use. All textual entailment tasks (including RTE and CB) are fully-held out during training of OPT-IML (Iyer et al., 2022).

## 4 Method

### 4.1 Models

We examine both pretrained-only and instruction-tuned models in this study. The former category is represented by LLaMA 30b (Touvron et al., 2023) and OPT 1.3b and 30b (Zhang et al., 2022). In

|  | prop. | prompt |
|---|---|---|
| **mood** | inter. | Do you find this movie review positive? |
|  | indic. | You find this movie review positive. |
|  | imper. | Tell me if you find this movie review positive. |
| **aspt.** | active | Do you find this movie review positive? |
|  | pass. | Is this movie review found positive? |
| **tense** | past | Did you find this movie review positive? |
|  | pres. | Do you find this movie review positive? |
|  | future | Will you find this movie review positive? |
| **modality** | can | Can you find this movie review positive? |
|  | could | Could you find this movie review positive? |
|  | may | May you find this movie review positive? |
|  | might | Might you find this movie review positive? |
|  | must | Must you find this movie review positive? |
|  | should | Should you find this movie review positive? |
|  | would | Would you find this movie review positive? |
| **synonymy** | apprai. | Do you find this movie appraisal positive? |
|  | comm. | Do you find this movie commentary positive? |
|  | criti. | Do you find this movie critique positive? |
|  | eval. | Do you find this movie evaluation positive? |
|  | review | Do you find this movie review positive? |

Table 1: Examples of variation of linguistic properties

the latter category, we consider OPT-IML 1.3b and 30b (Iyer et al., 2022)[4]. We make use of the HuggingFace transformers library (Wolf et al., 2020) for all our experiments.

### 4.2 Prompting setup

For our Sentiment Classification and NLI datasets, we map the label space to target words yes/no or yes/no/maybe in the case of two or three classes respectively. This was found to yield the best performance by Webson and Pavlick (2022) among alternative mappings. For question answering tasks, answer options are listed in a multiple-choice fashion using letters A, B or A, B, C, D which serve as target words. Additionally, we follow Gonen et al. (2022)'s advice regarding punctuation in prompts and add the postamble, e.g. *"Choices: yes or no? Answer:"* to every prompt, as this was found to aid zero-shot performance and reduce surface-form competition (Holtzman et al., 2021). Each prompt is evaluated on 500 random samples for each dataset. For datasets belonging to the same task, we keep the set of prompts fixed[5].

All our experiments are carried out in a zero-shot setting. Following Sanh et al. (2022); Webson and Pavlick (2022); Wei et al. (2021), we predict by recording which target word is assigned the largest log probability among all target words—regardless of whether that target word receives overall the

---

[4]To showcase similar performance variability for encoder-models, we include results on Flan-T5 (Chung et al., 2022) in Appendix C.

[5]For a full test sample with answer options see App A.

largest log probability across the entire vocabulary. We choose accuracy as our evaluation metric[6].

### 4.3 Linguistic variation in prompts

To contrast linguistic properties in prompts, we manually formulate parallel sets of prompts which differ in grammatical `mood`, `tense`, `aspect` or `modality`, systematically and one at a time. We present an example in Table 1 and the full list of prompts[7] for all tasks in Appendix D.

For example, in the case of `mood`, we design three sets of 10 prompts each which are identical, except that they are phrased either as a question, an order or a statement. The resulting sets of `interrogative`, `indicative` and `imperative` prompts are all in present tense and active voice, so as to guarantee a controlled setting. We proceed similarly for `aspect` by formulating two sets of `active` and `passive` prompts (all in interrogative mood, present tense) and `tense` by crafting prompts in `simple past`, `present` and `future` (all in active voice, interrogative mood). We also vary the degree of certainty in our instructions, while maintaining a minimal edit distance, by introducing different epistemic `modals` (example in Table 1). Here, all prompts are in active voice, present tense, interrogative mood. Lastly, we ask how model behaviour is linked to word sense ambiguity and word frequency. We thus replace content words in interrogative prompts with synonyms of varying word sense ambiguity and frequency.

**Statistical tests & analysis** We employ non-parametric tests, to quantify whether any observed performance variation between sets of prompts is indeed statistically significant. Specifically, we use the Friedman Two-Way Analysis of Variance by Ranks Test (Friedman, 1937) and the Wilcoxon Signed Rank Test (Wilcoxon, 1992) for paired samples. We investigate the influence of prompt length, perplexity, word sense ambiguity and frequency on accuracy by computing the Spearman Rank-Order Correlation Coefficient (Spearman, 1961) and the Pearson Correlation Coefficient (Pearson, 1895).

## 5 Results

### 5.1 Performance variability

Our results per task are presented in Tables 2, 3, 4.

---

[6] Run times and compute resources are detailed in App B.

[7] Our prompts are loosely inspired by prompts in Prompt-Source (Sanh et al., 2022). However, we restrict ourselves to simple sentence structures consisting only of one main clause.

**Mood.** As expected, LLMs generally respond more favourably to instructions phrased as questions or orders rather than statements, with the exception of LLaMA and OPT-IML 30b on BoolQ. However, we do not find that prompts in interrogative mood generally outperform imperative ones or vice versa. In most cases, instruction-tuning did not broaden the gap between indicative and interrogative/imperative prompts dramatically. Notably, we find cases where an individual indicative prompt performs best across all prompts (e.g. Table 9 details that 'This movie review makes people want to watch this movie.' achieves the highest accuracy of 97.6% for OPT-IML 1.3b on IMDB.).

**Aspect.** The hypothesis that `active` sentence constructions are simpler, shorter, more prevalent in the data and thus yield better performance was generally not confirmed. No model shows a clear preference for instructions phrased in active vs. passive voice across all datasets, with the exception of OPT-IML 1.3b for which active voice works consistently better, albeit not always significantly. Interestingly, for all other models, passive prompts generally yield better results on BoolQ. For our 30b models, we find active prompts to be superior only for RTE and ARC-E (Tables 3, 4).

**Tense.** Similarly, the hypothesis that prompts in `present` tense perform best, since they cater in particular to prompts seen during instruction-tuning, was only confirmed for CB. (In two cases, `future` prompts performed on par or slightly better.) On the other datasets our results proved to be more mixed. None of the tenses outperforms others for SST and IMDB, across models. On occasion, future and past prompts considerably outperformed present prompts, e.g. for OPT-IML 30b on IMDB ($> 96\%$ acc. (past/future) vs. $89\%$ acc. (present)) or similarly for OPT 30b on SST. On BoolQ and ARC-E results varied less, with observed differences under 2.5 percentage points.

**Modality.** Replacing different modal verbs in an instruction also results in considerable performance variation with differences up to 10 and 17 points on SST for LLaMA on *'must'* vs. *'would'* or OPT 30b on *'may'* vs. *'might'* (Table 2). For OPT-IML 1.3b and 30b, such variation is reduced on SST with average accuracies falling within the range of $92.23\% - 93.03\%$ for both sizes. However, instruction-tuning doesn't preclude the possibility of significant performance variation also for larger

| | | LLaMA 30b | | OPT 1.3b | | OPT-IML 1.3b | | OPT 30b | | OPT-IML 30b | |
|---|---|---|---|---|---|---|---|---|---|---|---|
| | | SST | IMDB | SST | IMDB | SST | IMDB | SST | IMDB | SST | IMDB |
| mood | indicative | 82.53* | 91.58* | 64.98* | 71.88* | 64.3* | 91.92* | 76.2* | **69.97** | 91.07* | 71.33* |
| | interrogative | 83.35* | 89.79* | **87.98** | 58.38* | **92.8** | 92.15* | **79.9** | 69.95 | 92.25* | **89.93** |
| | imperative | **83.65** | **92.42** | 76.85* | **74.08** | 92.25* | **92.92** | 79.08* | 66.98* | **92.68** | 87.5* |
| aspt. | active | **81.9** | 89.17 | 81.75* | **64.35** | **93.3** | **93.2** | 69.5* | **71.1** | 92.25* | 91.25* |
| | passive | 78.25* | **93.0** | **85.8** | 60.85* | 92.8 | 93.15 | **71.0** | 70.3 | **92.55** | **92.75** |
| tense | past | 80.8* | **94.67** | 71.05* | 80.53 | 86.95* | 94.85 | 85.8* | 82.66* | 91.68* | 96.12 |
| | present | 83.35 | 94.42 | **87.98** | 79.69* | **92.8** | 94.7 | 79.9* | **84.04** | **92.25** | 89.93* |
| | future | **85.72** | 93.0* | 76.6* | **80.79** | 88.37* | **94.88** | **87.35** | 83.79* | 92.18* | **96.15** |
| modality | can | 85.22* | 90.33* | 83.0* | 64.78* | 92.5* | 90.78* | 77.45* | 73.5* | 92.83 | **89.12** |
| | could | 84.75* | 91.38* | 77.47* | 67.12* | 92.42 | 90.2* | 70.97* | **74.02** | 92.68* | 87.58* |
| | may | 84.62* | 87.12* | 82.63* | 65.17* | 92.35 | 90.83* | 82.3* | 71.1* | 92.55* | 87.38* |
| | might | 83.85* | 91.25* | 77.18* | **69.72** | 92.25 | 91.1* | 65.98* | 72.7* | **92.92** | 85.58* |
| | must | 75.52* | 90.62* | 85.6* | 59.77* | 92.55* | **91.73** | **82.9** | 67.95* | 92.6* | 88.0* |
| | should | 82.92* | 91.33* | 85.47* | 63.08* | 92.78* | 90.05* | 81.32* | 70.15* | 92.73* | 88.25* |
| | would | **85.97** | **92.54** | **86.05** | 62.12* | **93.03** | 91.55 | 74.03* | 71.25* | 92.3* | 85.25* |
| synonymy | appraisal | 81.63* | **93.0** | 87.49* | **60.46** | 93.17* | 90.46* | 76.26* | **73.2** | **93.23** | 92.86 |
| | commentary | 81.6* | 92.95 | 86.37* | 60.23 | 92.91* | 88.03* | 64.34* | 71.86* | 92.97* | **93.09** |
| | critique | **84.4** | 92.29* | 85.66* | 58.94* | 92.46* | 91.11* | 68.63* | 71.46* | 92.43* | 91.8* |
| | evaluation | 83.63* | 92.0* | **89.89** | 52.34* | **93.29** | 91.74* | 78.26* | 70.23* | 92.49* | 91.26* |
| | review | 82.97* | 92.95 | 87.94* | 56.94* | 92.97* | **93.23** | **80.77** | 68.97* | 92.17* | 88.8* |
| | null prompt | 83.2 | 72.8 | 41.2 | 64.2 | 12.8 | 93.0 | 65.8 | 74.8 | 37.0 | 86.2 |
| | chance | 50 | 50 | 50 | 50 | 50 | 50 | 50 | 50 | 50 | 50 |

Table 2: Average accuracy per prompt in categories mood, aspect, tense, modality, synonymy on SST and IMDB (Sentiment Classification). Highest accuracy per category for each model and dataset marked in bold. Significant lower results per category marked with an asterisk. The null prompt contains no instruction.

| | | LLaMA 30b | | OPT 1.3b | | OPT-IML 1.3b | | OPT 30b | | OPT-IML 30b | |
|---|---|---|---|---|---|---|---|---|---|---|---|
| | | BoolQ | ARC-E | BoolQ | ARC-E | BoolQ | ARC-E | BoolQ | ARC-E | BoolQ | ARC-E |
| mood | indicative | **72.0** | 73.18 | 62.08 | 27.6 | 63.04 | 31.87 | 54.38* | 29.68* | **65.72** | 68.78* |
| | interrogative | 67.75* | **75.43** | 61.92 | **28.77** | **64.25** | 31.53 | 60.4 | **31.72** | 63.44* | 68.28 |
| | imperative | 64.58* | 74.65 | **62.72** | 27.52 | 63.7 | **34.1** | **60.52** | 30.47* | 64.81 | **69.16** |
| aspt. | active | 67.6* | **75.76** | 62.05* | **29.15** | **64.1** | **31.86** | 61.5 | **36.73** | 62.3 | **66.68** |
| | passive | **73.9** | 72.98* | **62.55** | 28.14 | 63.55* | 30.05* | **61.55** | 32.41* | **63.3** | 64.17* |
| tense | past | 66.83* | **75.0** | 59.28* | 28.02* | 49.15 | 29.23* | **63.69** | **28.4** | 66.35 | 70.96* |
| | present | 67.03* | 72.68* | 59.13* | **28.19** | **49.5** | 30.2* | 63.3* | 27.92 | **67.13** | 71.11* |
| | future | **67.43** | 73.03* | **59.61** | 27.93* | 49.07* | **30.3** | 63.03* | 28.15* | 66.91* | **71.19** |
| modality | can | 64.5* | 73.82* | 61.75* | 28.02* | 63.62 | 31.41* | **62.13** | 34.42 | 63.75* | 67.22* |
| | could | 63.75 | 73.82* | 61.88 | 28.27* | 63.25 | 31.28* | 61.38 | 33.92 | 63.5* | 67.47* |
| | may | 62.62* | 74.21* | **62.13** | 28.27 | 64.0 | **31.78** | 60.5* | 32.41* | 63.5* | 67.89 |
| | might | 63.5 | 75.52* | 61.88 | 28.39* | 63.88* | 31.41 | 60.38* | 32.79* | 62.5* | 67.73 |
| | must | 65.0* | 75.79* | 62.0 | 28.27* | 63.5* | 30.53* | 57.38* | 29.27* | **65.38** | 67.64* |
| | should | **67.75** | 75.65 | 61.87 | **28.77** | **64.13** | 30.4* | 58.88* | 29.77* | 63.88* | **68.9** |
| | would | 67.12 | **77.09** | 61.5 | 28.39* | 64.0 | 30.03* | 60.62* | **34.67** | 64.0* | 68.23* |
| synonymy | proper | 66.02* | 75.93 | **62.72** | 27.32 | 63.76* | 33.56* | **61.68** | 32.13 | 63.9 | 68.67 |
| | right | 62.76* | 76.1 | 62.64 | 27.1* | 63.94* | **34.08** | 61.12* | 31.77* | 64.44* | 68.11* |
| | correct | 62.9* | **76.16** | 62.28* | 27.65* | 63.78* | 33.82 | 60.3* | 31.73* | 64.24* | 68.21* |
| | appropriate | **67.24** | 75.85* | 62.64* | **27.73** | **63.98** | 33.34* | 61.32* | **32.31** | **64.5** | **68.91** |
| synonymy | answer | 61.82* | 76.18 | 62.33* | 27.71* | 63.82* | **33.69** | 60.07* | **31.94** | 64.15* | **68.47** |
| | reply | 62.93* | 75.43* | 62.22* | 27.73* | **63.91** | 33.42 | 60.73* | 31.28* | **64.64** | 67.81 |
| | response | **65.98** | **76.39** | 62.35* | 27.35* | 63.85 | 32.94* | **61.45** | 31.64* | 64.31* | 68.3 |
| | solution | 62.73* | 73.11* | **63.02** | **28.02** | 63.89* | 33.14* | 60.6 | 30.89* | 64.33* | 67.55* |
| | null prompt | 64.0 | 75.0 | 61.5 | 26.13 | 68.0 | 29.29 | 68.0 | 28.28 | 72.0 | 63.64 |
| | chance | 50 | 25 | 50 | 25 | 50 | 25 | 50 | 25 | 50 | 25 |

Table 3: Average accuracy per prompt in categories mood, aspect, tense, modality, synonymy on BoolQ, ARC-E (Question Answering). Highest accuracy per category for each model and dataset marked in bold. Significant lower results per category marked with an asterisk. The null prompt contains no instruction.

models of eg. 4 percentage points of OPT-IML 30b on IMDB and CB. On QA (Table 3) and NLI (Table 4) datasets, we find numerous examples of drops in accuracy by up to 5 percentage points.

**Synonymy.** Perhaps most surprisingly, replacing content words with non-standard synonyms does not generally hurt performance, but rather improves it. In particular, for SST and IMDB the content

| | | LLaMA 30b | | OPT 1.3b | | OPT-IML 1.3b | | OPT 30b | | OPT-IML 30b | |
|---|---|---|---|---|---|---|---|---|---|---|---|
| | | RTE | CB | RTE | CB | RTE | CB | RTE | CB | RTE | CB |
| mood | indicative | 53.28* | 49.7* | 47.9* | 51.9* | 58.55* | 62.27* | 51.18* | 46.77* | 67.96* | 79.67 |
| | interrogative | 57.6* | **53.53** | **48.0** | **58.17** | 58.95* | 62.33 | 51.04* | **59.33** | 69.58* | 75.47* |
| | imperative | **57.68** | 52.4 | 47.35* | 52.13* | **60.4** | 61.4 | **52.4** | 59.27* | **70.22** | **80.1** |
| aspt. | active | **60.62** | **53.68** | 52.9* | 56.28* | **55.65** | 59.88 | 52.38 | 58.56 | 69.5 | 70.84 |
| | passive | 60.46* | 53.08* | **53.0** | **57.0** | 54.95 | 59.88* | 51.8* | 55.56* | 69.16 | 68.56* |
| tense | past | **53.11** | 51.0* | **52.15** | 58.0 | 65.63* | 62.17* | **52.15** | 60.3 | 70.7* | 72.13* |
| | present | 52.96* | **51.63** | 51.16* | **58.73** | 65.9 | 62.43* | 52.08 | **60.53** | **71.78** | **74.23** |
| | future | 52.51 | 48.93* | 52.1 | 55.87* | 65.2* | **62.63** | 51.43* | 60.53 | 70.72 | 71.58* |
| modality | can | 60.55* | 53.07* | 54.04 | 56.6* | 56.12 | 61.83* | **53.17** | 59.07* | 71.65* | 74.53* |
| | could | 61.37* | **53.37** | 53.17* | **58.2** | 56.38 | 62.37* | 51.5* | 58.97* | 70.7* | 74.33* |
| | may | 60.18* | 52.27* | **54.75** | 58.07* | 54.58* | **64.03** | 52.28* | 56.6* | 72.17* | 73.6* |
| | might | 60.37* | 51.87* | 53.5* | 57.03* | 54.29* | 63.13* | 51.53* | 56.43* | **72.2** | 72.3* |
| | must | 57.25* | 50.43 | 54.25* | 54.93* | 52.92* | 61.73 | 51.23* | 56.33* | 70.62 | 70.53* |
| | should | 58.38* | 49.57* | 54.04* | 55.93* | 55.67 | 60.7* | 51.07* | 60.73 | 71.55 | 71.3* |
| | would | **62.75** | 51.63* | 53.75* | 56.03* | 54.17* | 60.43* | 51.55* | 60.17* | 71.0* | **74.87** |
| syn. | assertion | **57.1** | **52.0** | **50.8** | 58.6 | 66.8* | 58.6 | **53.4** | 67.8 | 74.3 | 78.2 |
| | claim | 55.5* | 49.6* | 49.6 | 58.6 | 68.0 | 60.6 | 51.9* | 66.0* | **74.4** | **78.8** |
| syn. | entailment | **55.4** | 42.4* | 50.1* | 55.4* | 63.6 | 60.4* | 49.0* | 46.0 | **70.2** | **75.2** |
| | implication | 54.8* | **54.4** | **51.0** | **60.2** | 64.1 | 63.8 | 50.0 | 46.4 | 69.9 | 75.0 |
| | null prompt | 45.0 | 55.2 | 40.0 | 57.6 | 46.5 | 61.6 | 46.6 | 66.8 | 41.2 | 79.2 |
| | chance | 50 | 33.3 | 50 | 33.3 | 50 | 33.3 | 50 | 33.3 | 50 | 33.3 |

Table 4: Average accuracy per prompt in categories mood, aspect, modal verbs, synonymy on RTE, CB (Question Answering). Highest accuracy per category for each model and dataset marked in bold. Significant lower results per category marked with an asterisk.

word *'review'* does not guarantee optimal performance (Table 2). This holds even for OPT-IML 30b which has been trained on instructions from FLAN and PromptSource most of which contain the word *'review'*. Instead rare synonyms such as *'appraisal'* and *'commentary'* yield better performance. Similarly, on BoolQ and ARC-E we did not find that prompts containing the words *'correct'* and *'answer'* worked best—even if many of the prompts in FLAN and PromptSource contain those words and none of the other synonyms we tested. Notably often, models respond more favourably to the rarer synonym *'appropriate'* (See Table 3).

## 5.2 Prompt transfer

When evaluating on a fixed prompt for different models and datasets, one tacitly assumes that prompts are to some extent 'universal'. Our results largely contradict this assumption (Tables 9, 11, 13). We found numerous cases in which a prompt performed optimally for one model on one dataset, but gave staggeringly poor results on other datasets. For instance, the best prompt for OPT-IML 1.3b on IMDB yields 97.6% accuracy, but barely above chance performance on SST. Similarly, we saw large drops in performance when transferring optimal prompts from IMDB to SST and vice versa for LLaMA 30b, OPT 1.3b and OPT 30b.

Prompts that were optimal for one model and dataset also transferred poorly to other models. We found many cases of drops by more than 20 percentage points on SST and IMDB or 5 percentage points on BoolQ and CB.

## 5.3 The relation between robustness and instruction-tuning

Instruction-tuning holds promise of improved performance *and* robustness, but how robust can we expect our results to be? In line with previous works, we find our instruction-tuned models to perform more reliably on seen tasks than their pre-trained counterparts of the same size. For OPT 30b, accuracy can vary by 10 points or more for SST and IMDB. For OPT-IML 30b this gap narrows, but remains non-negligible. While performance on SST stabilises between 91% and 93% for SST, we still see performance in the range 85 − 92% on IMDB (Table 2). ARC-E is not part of the instruction-tuning tasks for OPT-IML 30b and performance here varies by 5 percentage points (Table 3). Similarly, performance on RTE and CB varies significantly with accuracies in the ranges 67.96 − 72.2% and 68.56 − 80.1% respectively (Table 4).

## 5.4 The relation between robustness and model size

We find numerous examples of increased model size not leading to increased stability. For instance,

changing *'must'* to *'might'* results in a performance drop by 17 percentage points for OPT 30b on SST (Table 2). Overall, when comparing OPT-IML with OPT at 1.3b and 30b, the gap between best and worst prompts closes only for RTE, is stable for BoolQ, ARC-E and SST and widens for IMDB, and CB e.g. from 5 to 12pp (Tables 9, 11, 13).

## 6 Analysis

Since accuracy varies considerably, we now analyse whether higher accuracy can be explained by lower prompt perplexity, prompt length, or the use of less ambiguous or more frequent words in the prompt. We present correlation results in Table 5.

**Prompt perplexity** Following Gonen et al. (2022), we average across perplexity for 500 random test samples each accompanied by the instruction in question. We find that perplexity scores often reflect linguistic intuition, e.g. they are lower for prompts in imperative vs. indicative mood or prompts containing *'answer'* vs. other synonyms (see Appendix E). Surprisingly however, we do not find lower perplexity to correlate significantly with higher accuracy across models or datasets (Table 5). For LLaMA 30b, *higher* perplexity correlates with higher accuracy (except on IMDB and BoolQ). OPT-IML 30b performs better given higher perplexity prompts. Overall, our findings contradict Gonen et al. (2022) and indicate that the success of high or low perplexity prompts is particular to any combination of dataset and model.

**Frequency of synonyms** We approximate word frequency as the number of occurrences in the 14b Intelligent Web-based Corpus (Davies and Kim, 2019). In general, we do not find that frequent synonyms lead to better performance (Table 5). For BoolQ and ARC-E, correlation between word frequency and accuracy oscillates around zero with no clear preference for frequent synonyms emerging. For SST and IMDB, infrequent synonyms tend to work better for OPT IML 30b. For OPT, correlation is positive for SST and negative for IMDB's longer sentences. For LLaMA correlation coefficients are mostly around zero. On RTE and CB, more frequent synonyms lead to better performance except for LLaMA and OPT 1.3b on RTE.

**Ambiguity of synonyms** We quantify word sense ambiguity as the number of word senses in WordNet (Miller, 1995). Intuitively, one would expect more ambiguous synonyms to pose greater

difficulty for LLMs and to lower accuracy. We largely do not find this to be the case (Table 5). While correlation between accuracy and degree of ambiguity is generally around zero for BoolQ and ARC-E, on CB it is positive for all models. For SST and IMDB, OPT-IML 30b performs better on less ambiguous prompts, potentially due to its size, while the opposite is true of OPT-IML 1.3b.

**Prompt length** Overall, none of our models perform better on longer or shorter prompts (in number of tokens) across all tasks (Table 5). For LLaMA 30b and OPT 1.3b, longer prompts result in significantly higher accuracies only for IMDB and BoolQ. Performance of OPT-IML 1.3b correlates positively with prompt length (except on SST and CB). For OPT-IML 30b results are mixed.

## 7 Lessons learnt and way forward

### 7.1 Implications of our research

**Instability in prompting** Our findings clearly demonstrate the instability of prompt-based evaluation (§5) that is rarely featured in performance reports. For any model and task, differences in performance can be considerable at even the slightest change in wording or sentence structure.

**The connection between data distribution and model behaviour** Our findings should be taken as an invitation to revisit the common assumption that LLMs respond best to lower perplexity prompts containing simple and frequent words and grammatical structures. We find numerous cases where LLMs do not learn from the data distribution in the way one would assume based on perplexity scores and linguistic intuition (§6). This calls for further investigation into the interplay between model behaviour and the distribution of language use during pretraining and instruction-tuning.

**The effect of instruction-tuning** Contrary to prevailing opinion, the above holds also for comparatively larger instruction-tuned LLMs. Our results indicate that instruction-tuning should not be taken as a panacea to performance instability without further investigation (§5.3). While it does overall improve performance and robustness, performance can still vary by over 5pp on seen tasks.

**Limitations of current evaluation practices** Importantly, our work highlights the limitations of benchmarking LLMs on the same prompt for a given task, or only a small set of prompts, which

| | task | LLaMA 30b $\rho_s$ | $\rho_p$ | OPT 1.3b $\rho_s$ | $\rho_p$ | OPT-IML 1.3b $\rho_s$ | $\rho_p$ | OPT 30b $\rho_s$ | $\rho_p$ | OPT-IML 30b $\rho_s$ | $\rho_p$ |
|---|---|---|---|---|---|---|---|---|---|---|---|
| ppl. vs acc. | SST | 0.23* | 0.3* | 0.27* | 0.16 | 0.49* | 0.22* | 0.11 | 0.21* | −0.02 | 0 |
| | IMDB | −0.17* | −0.07 | −0.06 | 0.04 | −0.06 | 0.02 | −0.37* | −0.18* | 0.18* | 0.13 |
| | BoolQ | −0.02 | 0.02 | −0.21* | −0.3* | −0.04 | −0.17* | 0.22* | 0.17* | 0.13 | 0.06 |
| | ARC-E | 0.2* | 0.16* | −0.07 | −0.04 | −0.14* | −0.15* | −0.12 | −0.12 | 0.11 | 0.07 |
| | RTE | 0.15 | 0.16* | 0.08 | 0.08 | −0.07 | −0.0 | −0.28* | −0.13 | 0.05 | 0.51* |
| | CB | 0.46* | 0.17* | 0.43* | 0.29* | 0.15* | 0.58* | −0.59* | −0.47* | −0.12 | 0.42* |
| amb. vs acc. | SST | 0.06 | 0.01 | 0.23 | 0.07 | 0.1 | 0.03 | 0.35* | 0.21 | −0.17 | −0.3 |
| | IMDB | 0.08 | 0.09 | −0.07 | −0.09 | 0.53* | 0.53* | −0.08 | −0.12 | −0.35* | −0.57* |
| | BoolQ | −0.11 | −0.09 | 0.06 | 0.04 | −0 | 0.02 | −0.02 | 0.03 | −0.04 | −0.01 |
| | ARC-E | 0.06 | 0.05 | −0.16 | −0.21 | −0.03 | 0.09 | −0.1 | −0.03 | −0.14 | −0.09 |
| | CB | 0.12 | 0.4 | 0.1 | 0.25 | 0.22 | 0.39 | 0.41 | 0.21 | 0.05 | 0.08 |
| | RTE | −0.24 | −0.36 | −0.2 | −0.14 | 0.59 | 0.43 | 0.39 | 0.15 | 0.27 | 0.14 |
| freq. vs acc. | SST | 0.09 | 0.02 | 0.2 | 0.1 | 0.05 | 0.05 | 0.11 | 0.19 | −0.43* | −0.35* |
| | IMDB | −0.03 | 0.07 | −0.43* | −0.18 | 0.39* | 0.5* | −0.18 | −0.14 | −0.47* | −0.6* |
| | BoolQ | −0.11 | −0.08 | 0.06 | 0.05 | 0.05 | 0.06 | −0.08 | 0.03 | −0.01 | 0.04 |
| | ARC-E | 0.02 | 0.05 | −0.05 | −0.19 | −0.03 | 0.12 | −0.02 | 0.02 | −0.04 | −0.04 |
| | CB | 0.12 | 0.4 | 0.1 | 0.25 | 0.22 | 0.39 | 0.41 | 0.21 | 0.05 | 0.08 |
| | RTE | −0.24 | −0.36 | −0.2 | −0.14 | 0.59 | 0.43 | 0.39 | 0.15 | 0.27 | 0.14 |
| len. vs acc. | SST | −0.23* | −0.37* | −0.27* | −0.19 | −0.16 | 0.06 | −0.14 | −0.27* | 0.14 | 0.23* |
| | IMDB | 0.15 | 0.05 | 0.16 | 0.19 | 0.23* | 0.39* | 0.54* | 0.47* | −0.14 | 0.03 |
| | BoolQ | 0.11 | 0.06 | 0.1 | 0.2 | −0.09 | 0.18 | 0.16 | 0.19 | −0.27* | −0.26* |
| | ARC-E | −0.14 | −0.13 | −0.0 | −0.02 | 0.27* | 0.32* | 0.24* | 0.25* | −0.15 | −0.08 |
| | RTE | −0.47* | −0.45* | −0.46* | −0.55* | 0.13 | 0.11 | −0.08 | −0.17 | −0.31* | −0.1 |
| | CB | −0.38* | −0.33* | −0.4* | −0.41* | −0.27* | −0.19* | 0.03 | 0.22* | 0.43* | 0.35* |

Table 5: Spearman ($\rho_s$) and Pearson correlation coefficients ($\rho_p$) of (1) perplexity, (2)word sense ambiguity, (3) frequency of synonyms, and (4) prompt length against accuracy. Significant results ($p < 0.05$) marked with an asterisk.

is the current practice. Large variability in performance (§5) and a lack of transparency around used prompts make evaluations unreliable and hard to reproduce. Individual prompts are not transferable across datasets or models (§5.2). Instruction-tuning does not appear to solve these problems (§5.3).

## 7.2 Recommendations

Based on the lessons learnt in our study, we put forward a proposal for a more robust and comprehensive evaluation framework for LLM prompting.

**Collect prompts that represent linguistic variability.** In order to obtain a robust and accurate estimate of model performance, one needs to account for linguistic variability of prompts, ideally in a controlled and rigorous manner. This can be accomplished by collecting sets of candidate prompts that are representative of core linguistic structures.

*Use semi-automatic approaches such as controlled paraphrasing* (Iyyer et al., 2018) to generate prompts that vary in grammatical constructions in a systematic way.

*Replace content words with synonyms* to diversify vocabulary. Our results on synonyms point to a simple method for expanding prompt sets to cover a more diverse vocabulary. This enables controlled investigation of the the link between word sense ambiguity, frequency, and model behaviour.

**Generate a sufficiently large set of prompts.** Given the high levels of performance variability between different prompts (up to 17 pp, as observed in our experiments), it is crucial to experiment with a sufficiently large set of prompts, so as to obtain statistically reliable estimates of performance, independent of individual prompt formulations.

*Include estimates of performance mean* and variance based on a large set of prompts for a more accurate picture of model capabilities.

*Treat prompts as hyperparameters.* When choosing a single prompt for evaluation, select it per model and dataset on a held-out development set.

**Standardize and report metrics** characterising the prompt set and its impact on performance.

*For prompt collections, report metrics* such as perplexity, degree of ambiguity, and their distribution. Analyse how these correlate with performance metrics or log probabilities of true labels using correlation coefficients (Gonen et al., 2022).

*Use mixed effects models* (Gelman and Hill, 2006) to analyse how prompt characteristics influence model performance per dataset and sample (Lampinen et al., 2022).

## 8 Conclusion

We evaluated five LLMs, both pretrained and instruction-tuned, on parallel sets of prompts that systematically differ in grammatical mood, aspect, tense, modality and use of synonyms. We find that there is no favoured sentence structure that performs best across models and tasks. Prompts generally transfer poorly across datasets and models. We find considerable performance variation, which can still persist even for larger instruction-tuned LLMs on seen tasks and is not explained by perplexity, word sense ambiguity or word frequency.

## Limitations

In this work, we experiment with LLMs of up to 30b parameters that are able to perform a variety of tasks in a zero-shot fashion. We did not include larger open-source LMs or OpenAI's GPT-3 due to (computational) cost. Further at the time of writing, the OpenAI API provides only the top 5 log probabilities given any input. This stands potentially at odds with the evaluation procedure of making a prediction based on which answer option receives the highest log probability, independently of whether that answer option occurs in the top 5 log probabilities. We did not include results for smaller variants of OPT (IML) and LLaMA, since they did not perform significantly above chance across all our tasks in the zero-shot setting in initial experiments.

So as to not introduce an additional source of variation in our experiments and observe the effect of linguistic variation in as much isolation as possible, we did not include experiments on in-context learning or priming (Lu et al., 2022; Zhou et al., 2022; Zhao et al., 2021; Min et al., 2022). We focus on the zero-shot setting only, so that models can infer a given task only based on an instruction without any demonstrations.

Future experiments could include more experiments on other architectures such as encoder-decoder models, e.g. T5 (Raffel et al., 2020), T0 (Sanh et al., 2022) or Flan-T5 (Chung et al., 2022), or multilingual models, e.g. bloom (Scao et al., 2022) or bloomz (Muennighoff et al., 2022). Investigating model behaviour based on linguistic properties in languages that are morphologically richer than English would equally pose an interesting avenue for further research.

In future research, we would also like to draw a comparison with an instruction-tuned version of LLaMA. At the time of writing, we are only aware of models such as Vicuna[8] and Alpaca[9] which are trained on data generated by OpenAI text-davinci-003 or interactions with human users, unlike OPT-IML which has been fine-tuned on a large collection of NLP tasks. We did thus not include Vicuna and Alpaca in our experiments so as to avoid a skewed comparison.

## Ethics & broader impact

In this work, we analyse LLM behaviour given a variety of linguistic properties provided to them as prompts. We uncover that models process language which varies in word sense ambiguity, frequency, perplexity, and length of expression in unexpected ways. Based on our findings, we provide recommendations for reliable and robust evaluation practices. Providing recommendations for prompt engineering from a linguistic point of view is not the decided aim of this study and indeed any recommendations that could be derived from our results appear localised to the context of a particular models or dataset. We publicly release our set of 550 prompts in Appendix D, as a basis for further research on LLM behaviour through the lens of linguistics.

While our work does not include tasks close to real-world applications such as hate speech detection, our findings indicate that performance on such tasks might also vary considerably under different instructions. We thus advise NLP practitioners working on sensitive applications with very large LMs to carefully evaluate model performance across a broader range of semantically equivalent instructions. In particular, evaluation reports should include measures of performance variability across prompts, seeds and demonstration examples. Further, prompts that have been engineered based on a particular model and dataset should not be transferred to other datasets or domains, since in general universality of optimal prompts cannot be assumed. Developers of LLMs for hate speech detection and related tasks should account for instabilities, particularly when using LLMs for automatic annotation.

---

[8] https://github.com/lm-sys/FastChat
[9] https://crfm.stanford.edu/2023/03/13/alpaca.html

## Acknowledgements

We thank our anonymous reviewers for their insightful comments. The work for this publication is financially supported by the project, 'From Learning to Meaning: A new approach to Generic Sentences and Implicit Biases' (project number 406.18.TW.007) of the research programme SGW Open Competition, which is (partly) financed by the Dutch Research Council (NWO).

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

## A Example prompt

In this section, we give an example prompt for sentiment classification featuring a random input sample from SST-2.

---
Prompt:
*Do you want to watch this movie based on this movie review?*

*An entertaining British hybrid of comedy, caper thrills and quirky romance.*
*Choices: yes or no? Answer:*

Answer choices:
*yes, no*

True answer:
*yes*

---

## B Average runtime

On average, evaluating LLaMA, OPT or OPT-IML 30b on all prompts for one dataset took approximately two hours using four NVIDIA A100 GPUs. Evaluating OPT or OPT-IML 1.3b on all prompts for one dataset took around 15 minutes using one GPU.

## C Performance variability in encoder-decoder models

In preliminary experiments we also considered encoder-decoder models, e.g. Flan-T5 (Chung et al., 2022). We later restricted our experimental setup to decoder-only models, since they are more widely used to-date. To demonstrate performance variability similar to the main results in the paper also for an encoder-decoder model we include illustrative results on mood and synonymy for Flan-T5 (XL) for sentiment classification and NLI in Tables 6, 7, 8.

## D Performance per prompt

We list all prompts used in our experiments for sentiment analysis (Table 9), NLI (Table 13) and Question Answering (Table 11) grouped by the grammatical properties we investigate. Properties we investigate are mood (indicative, imperative, interrogative), aspect (active, passive), tense (past, present, future), modality (can, could, may, might, must, should, would), and synonymy (different synonyms per task). Note that the number of prompts

|  |  | SST | IMDB | RTE | HANS |
|---|---|---|---|---|---|
| mood | ind. | 86.6 | 94.6 | 86.1 | 76.4 |
|  | inter. | 87.3 | 95.8 | 88.8 | **77.5** |
|  | imp. | **88.2** | **96.1** | **90.9** | 77.3 |

Table 6: Average accuracy per prompt in category mood (indicative, interrogative, imperative) for Flan-T5 XL on SST-2, IMDB, RTE, HANS. Highest accuracy per dataset marked in bold.

|  |  | RTE | HANS |
|---|---|---|---|
| syn. | entailment | 95 | 77 |
|  | implication | **96** | **78.4** |
| syn. | assertion | 91 | **81.5** |
|  | claim | **92.5** | 81 |

Table 7: Average accuracy per prompt in category synonymy for Flan-T5 XL on RTE, HANS. Highest accuracy per dataset marked in bold.

|  |  | SST | IMDB |
|---|---|---|---|
| synonymy | appraisal | 83.4 | 85.7 |
|  | commentary | 77.7 | 80.4 |
|  | critique | 81.5 | 85.8 |
|  | evaluation | **83.7** | **88.4** |

Table 8: Average accuracy per prompt in category synonymy for Flan-T5 XL on SST-2, IMDB. Highest accuracy per dataset marked in bold.

per property, eg. for mood vs. aspect, might differ, since not all prompts listed under mood (Eg., *'Is this a good movie?'*) can be phrased in active and passive voice. They are thus omitted from active and passive. We detail accuracy per prompt for all models used in our experiments.

## E Perplexity per category

In Tables 15, 16, 17, 18, 19, 20 we list perplexity scores for each linguistic property and model in each task.

| property | prompt | LLaMA 30b | | OPT 1.3b | | OPT-IML 1.3b | | OPT 30b | | OPT-IML 30b | |
|---|---|---|---|---|---|---|---|---|---|---|---|
| | | SST | IMDB | SST | IMDB | SST | IMDB | SST | IMDB | SST | IMDB |
| - | \<null prompt\> | 83.2 | 72.8 | 41.2 | 64.2 | 12.8 | 93.0 | 65.8 | 74.8 | 37.0 | 86.2 |
| ind. | You find this movie review positive | 84.2 | 94.0 | 89.4 | 62.0 | 91.4 | 89.2 | 87.4 | 66.2 | 91.4 | 59.6 |
| ind. | This is a good movie | 82.4 | 88.33 | 47.0 | 77.2 | 35.8 | 79.4 | 53.2 | 70.6 | 89.0 | 66.8 |
| ind. | The label for this movie review is positive | 84.2 | 92.67 | 58.8 | 75.6 | 92.0 | 92.2 | 81.8 | 66.6 | 90.6 | 60.8 |
| ind. | You like the movie based on this movie review | 85.0 | 91.67 | 83.6 | 64.6 | 89.2 | 92.6 | 82.0 | 70.0 | 91.8 | 73.4 |
| ind. | This movie review makes people think this is a good movie | 85.6 | 91.33 | 43.8 | 77.2 | 46.4 | 97.2 | 69.4 | 75.0 | 93.0 | 83.6 |
| ind. | This movie review makes people want to watch this movie | 79.8 | 91.0 | 54.8 | 78.4 | 51.6 | **97.6** | 85.4 | 69.0 | 91.0 | 78.2 |
| ind. | This movie review is positive | 80.6 | 92.67 | 58.0 | 75.8 | 63.0 | 92.0 | 80.4 | 67.0 | 90.4 | 71.4 |
| ind. | You want to watch this movie based on this movie review | 78.4 | 91.0 | 84.4 | 64.2 | 45.0 | 95.2 | 70.0 | 75.4 | 91.4 | 76.8 |
| inter. | Do you find this movie review positive? | 90.2 | 90.3 | 89.8 | 49.8 | 93.6 | 94.6 | 89.2 | 56.6 | 91.2 | 85.4 |
| inter. | Is this a good movie | 88.0 | 93.3 | 88.2 | 64.2 | 91.6 | 80.8 | 73.8 | 73.2 | 92.8 | 91.6 |
| inter. | Is the label for this movie review positive | 86.0 | 90.0 | 90.4 | 59.6 | 92.8 | 95.4 | 85.6 | 61.2 | 92.6 | 93.8 |
| inter. | Do you like the movie based on this movie review | 85.8 | 89.3 | 89.2 | 59.6 | 93.0 | 90.8 | 88.2 | 73.4 | 92.2 | 88.2 |
| inter. | Does this movie review make people think this is a good movie | 80.4 | 85.0 | 79.2 | 63.0 | 92.8 | 94.4 | 45.4 | 78.6 | 92.2 | 91.0 |
| inter. | Does this movie review make people want to watch this movie | 87.0 | 94.0 | 88.4 | 60.6 | 92.8 | 93.4 | 80.0 | 78.8 | 92.4 | 90.0 |
| inter. | Is this movie review positive | 88.0 | 90.7 | 90.2 | 54.0 | **93.8** | 94.6 | 89.0 | 61.2 | 92.2 | 92.4 |
| inter. | Do you want to watch this movie based on this movie review | 87.0 | 85.7 | 88.4 | 56.2 | 92.0 | 93.2 | 88.0 | 76.6 | 92.4 | 87.0 |
| imp. | Tell me if you find this movie review positive | 86.8 | 94.67 | 85.8 | 67.2 | 93.2 | 94.0 | 86.6 | 60.2 | 91.8 | 82.2 |
| imp. | Tell me if this is a good movie | 87.4 | 94.0 | 73.6 | 73.6 | 92.2 | 84.8 | 73.0 | 77.2 | 92.6 | 88.0 |
| imp. | Tell me if the label for this movie review is positive | 90.0 | 92.33 | 76.8 | 73.8 | 92.6 | 94.6 | 87.0 | 61.4 | 93.2 | 91.0 |
| imp. | Tell me if you like the movie based on this movie review | 83.8 | **95.0** | 80.0 | 71.0 | 92.0 | 94.2 | 87.8 | 69.4 | 92.6 | 85.6 |
| imp. | Tell me if this movie review makes people think this is a good movie | 70.2 | 83.67 | 68.0 | **81.4** | 91.8 | 93.8 | 39.0 | 76.0 | 93.6 | 90.8 |
| imp. | Tell me if this movie review makes people want to watch this movie | 86.8 | 93.67 | 75.2 | 78.6 | 92.8 | 93.0 | 86.6 | 70.6 | 92.8 | 90.8 |
| imp. | Tell me if this movie review is positive | 87.4 | 93.0 | 80.8 | 71.6 | 93.2 | 94.6 | **89.4** | 45.6 | 92.4 | 88.4 |
| imp. | Tell me if you want to watch this movie based on this movie review | 76.8 | 93.0 | 74.6 | 75.4 | 90.2 | 94.4 | 83.2 | 75.4 | 92.4 | 83.2 |
| active | Do people consider this movie review positive | 87.4 | 91.67 | 88.4 | 60.4 | 93.6 | 93.6 | 68.6 | 68.8 | 92.0 | 91.2 |
| active | Do people label this movie review as positive | 87.8 | 89.33 | 80.0 | 64.8 | 93.6 | 95.0 | 80.2 | 64.4 | 91.6 | 90.4 |
| active | Do people like this movie | 87.4 | 93.67 | 81.0 | 68.0 | 93.0 | 90.2 | 75.2 | 73.4 | 93.0 | 93.0 |
| active | Does this movie review make people think that this is a good movie | 65.0 | 82.0 | 77.6 | 64.2 | 93.0 | 94.0 | 54.0 | 77.8 | 92.4 | 90.4 |
| passive | Is this movie review considered positive by people | 88.8 | 94.0 | 87.6 | 59.2 | 93.4 | 94.4 | 67.0 | 67.2 | 92.0 | 91.6 |
| passive | Is this movie review labelled as positive by people | 85.4 | 94.33 | 88.0 | 60.2 | 93.2 | 94.4 | 69.2 | 69.6 | 92.2 | 91.6 |
| passive | Is this movie liked by people | 85.4 | 91.0 | 89.8 | 59.8 | 92.8 | 90.6 | 68.6 | 72.8 | 92.6 | 93.6 |
| passive | Are people made to think by this review that this is a good movie | 53.4 | 92.67 | 77.8 | 64.2 | 91.8 | 93.2 | 79.2 | 71.6 | 93.4 | 94.2 |
| can | Can you find this movie review positive | 83.2 | 94.67 | 87.6 | 56.0 | 93.0 | 93.8 | 87.6 | 67.2 | 92.4 | 89.33 |
| can | Can this be a good movie | 86.8 | 87.33 | 85.2 | 66.8 | 91.8 | 79.8 | 83.2 | 73.8 | 93.6 | 90.67 |
| can | Can the label for this movie review be positive | 88.4 | 93.33 | 82.6 | 70.6 | 92.4 | 93.2 | 83.0 | 68.2 | 93.2 | 91.0 |
| can | Can you like the movie based on this movie review | 86.6 | 91.33 | 88.6 | 65.2 | 92.8 | 92.6 | 87.4 | 72.8 | 92.2 | 87.67 |
| can | Can this movie review make people think this is a good movie | 79.4 | 89.67 | 65.8 | 67.4 | 92.4 | 91.0 | 36.6 | 81.6 | 93.4 | 89.67 |
| can | Can this movie review make people want to watch this movie | 84.2 | 85.33 | 82.6 | 68.0 | 93.0 | 91.8 | 75.8 | 79.2 | 93.0 | 88.0 |
| can | Can this movie review be positive | 85.4 | 89.67 | 83.2 | 68.0 | 92.8 | 91.0 | 79.8 | 70.8 | 92.8 | 90.33 |
| can | Can you want to watch this movie based on this movie review | 87.8 | 91.33 | 88.4 | 56.2 | 91.8 | 93.0 | 86.2 | 74.4 | 92.0 | 86.33 |
| could | Could you find this movie review positive | 87.2 | 93.33 | 82.6 | 59.2 | 93.0 | 93.4 | 89.8 | 62.4 | 92.4 | 88.67 |
| could | Could this be a good movie | 89.6 | 87.0 | 75.8 | 68.2 | 91.8 | 78.8 | 68.0 | 77.2 | 93.4 | 89.33 |
| could | Could the label for this movie review be positive | 86.4 | 94.67 | 82.4 | 68.4 | 93.0 | 90.4 | 78.4 | 70.0 | 93.2 | 90.67 |
| could | Could you like the movie based on this movie review | 86.4 | 91.67 | 85.4 | 66.2 | 93.0 | 92.8 | 82.2 | 73.8 | 92.0 | 84.33 |
| could | Could this movie review make people think this is a good movie | 67.6 | 93.67 | 55.4 | 70.0 | 92.8 | 90.4 | 29.8 | 80.6 | 93.2 | 89.0 |
| could | Could this movie review make people want to watch this movie | 85.0 | 88.0 | 74.0 | 71.8 | 92.8 | 91.4 | 63.8 | 80.6 | 92.8 | 87.67 |
| could | Could this movie review be positive | 87.0 | 89.0 | 76.0 | 68.0 | 92.8 | 91.2 | 75.2 | 71.2 | 92.4 | 90.33 |
| could | Could you want to watch this movie based on this movie review | 88.8 | 93.67 | 88.2 | 65.2 | 90.2 | 93.2 | 80.6 | 76.4 | 92.0 | 80.67 |
| may | May you find this movie review positive | 83.6 | 85.33 | 89.2 | 57.2 | 92.4 | 92.0 | 86.8 | 65.0 | 92.2 | 89.33 |
| may | May this be a good movie | 89.4 | 89.0 | 82.2 | 70.0 | 92.6 | 83.4 | 73.8 | 74.0 | 93.4 | 90.67 |
| may | May the label for this movie review be positive | 79.8 | 86.0 | 80.0 | 69.4 | 92.4 | 89.0 | 85.0 | 63.8 | 92.4 | 90.0 |
| may | May you like the movie based on this movie review | 81.8 | 86.0 | 84.6 | 62.6 | 92.6 | 93.0 | 88.0 | 73.2 | 92.0 | 84.67 |
| may | May this movie review make people think this is a good movie | 86.8 | 88.33 | 70.6 | 69.0 | 93.0 | 91.6 | 72.4 | 75.6 | 93.0 | 88.0 |
| may | May this movie review make people want to watch this movie | 76.8 | 81.67 | 83.6 | 67.8 | 93.6 | 92.6 | 90.0 | 72.8 | 92.4 | 87.0 |
| may | May this movie review be positive | 90.8 | 88.0 | 82.8 | 66.4 | 92.2 | 91.4 | 76.6 | 70.0 | 93.0 | 90.33 |
| may | May you want to watch this movie based on this movie review | 88.0 | 92.67 | 88.0 | 59.0 | 90.0 | 93.6 | 85.8 | 74.4 | 92.0 | 79.0 |
| might | Might you find this movie review positive | 87.8 | 90.67 | 87.6 | 62.6 | 93.2 | 93.8 | 84.8 | 66.4 | 92.0 | 90.33 |
| might | Might this be a good movie | 84.2 | 89.0 | 78.4 | 71.4 | 92.4 | 82.6 | 63.0 | 77.0 | 93.4 | 88.33 |
| might | Might the label for this movie review be positive | 89.4 | 93.0 | 82.4 | 69.6 | 92.6 | 92.0 | 53.4 | 69.0 | 92.8 | 91.0 |
| might | Might you like the movie based on this movie review | 87.0 | 91.0 | 80.8 | 67.2 | 91.2 | 91.8 | 76.8 | 73.8 | 92.8 | 75.33 |
| might | Might this movie review make people think this is a good movie | 62.0 | 95.0 | 58.8 | 76.2 | 92.4 | 92.2 | 30.6 | 76.4 | **94.4** | 89.0 |
| might | Might this movie review make people want to watch this movie | 86.2 | 87.67 | 76.6 | 73.2 | 93.6 | 93.0 | 85.8 | 73.8 | 92.8 | 88.0 |
| might | Might this movie review be positive | 85.6 | 90.67 | 75.4 | 68.6 | 93.0 | 91.4 | 62.0 | 69.6 | 92.8 | 90.33 |
| might | Might you want to watch this movie based on this movie review | 88.6 | 93.0 | 77.4 | 69.0 | 89.6 | 92.0 | 71.4 | 75.6 | 92.4 | 72.33 |
| must | Must you find this movie review positive | 81.8 | 89.33 | 89.8 | 51.4 | 92.6 | 95.6 | 89.0 | 58.2 | 92.2 | 90.67 |
| must | Must this be a good movie | 78.4 | 87.0 | 88.0 | 65.2 | 91.4 | 74.2 | 81.4 | 72.6 | 93.0 | 84.67 |
| must | Must the label for this movie review be positive | 62.6 | 95.0 | 82.8 | 69.2 | 92.8 | 94.0 | 86.0 | 56.8 | 93.0 | 90.33 |
| must | Must you like the movie based on this movie review | 73.6 | 94.33 | 90.2 | 59.4 | 93.0 | 92.6 | 89.2 | 70.0 | 92.4 | 86.67 |
| must | Must this movie review make people think this is a good movie | 63.2 | 91.0 | 71.8 | 64.6 | 92.8 | 95.0 | 62.0 | 75.6 | 93.0 | 89.0 |
| must | Must this movie review make people want to watch this movie | 89.8 | 86.33 | 84.6 | 61.8 | 93.0 | 94.2 | 76.6 | 73.8 | 92.6 | 87.33 |
| must | Must this movie review be positive | 66.6 | 90.0 | 88.6 | 54.8 | 93.4 | 94.2 | 80.4 | 64.8 | 92.6 | 89.67 |
| must | Must you want to watch this movie based on this movie review | 88.2 | 92.0 | 89.0 | 51.8 | 91.4 | 94.0 | 88.6 | 71.8 | 92.0 | 85.67 |
| should | Should you find this movie review positive | 89.6 | 92.0 | 88.8 | 58.0 | 93.6 | 93.8 | 87.2 | 68.4 | 92.0 | 88.67 |
| should | Should this be a good movie | 89.2 | 88.67 | 87.6 | 65.4 | 91.0 | 73.0 | 87.0 | 71.2 | 93.6 | 88.67 |
| should | Should the label for this movie review be positive | 88.6 | 94.0 | 86.6 | 65.8 | 93.2 | 93.2 | 88.8 | 56.4 | 93.4 | 91.0 |
| should | Should you like the movie based on this movie review | 85.8 | 91.33 | 82.6 | 64.8 | 93.0 | 92.2 | 85.2 | 73.8 | 92.2 | 88.67 |
| should | Should this movie review make people think this is a good movie | 57.8 | 94.33 | 73.2 | 69.0 | 92.0 | 91.4 | 42.4 | 79.6 | 93.0 | 88.33 |
| should | Should this movie review make people want to watch this movie | 82.6 | 87.33 | 88.2 | 63.4 | 93.2 | 90.2 | 84.2 | 78.8 | 93.0 | 87.0 |
| should | Should this movie review be positive | 82.4 | 91.33 | 89.8 | 57.2 | 93.4 | 93.4 | 88.6 | 57.0 | 92.6 | 86.67 |
| should | Should you want to watch this movie based on this movie review | 87.4 | 91.67 | 87.0 | 61.0 | 92.8 | 93.2 | 87.2 | 76.0 | 92.0 | 87.0 |
| would | Would you find this movie review positive | 89.6 | 95.33 | 90.6 | 51.6 | 93.2 | 94.2 | 89.4 | 57.8 | 91.6 | 80.33 |
| would | Would this be a good movie | 88.0 | 90.0 | 88.2 | 64.2 | 91.6 | 81.4 | 70.0 | 76.8 | 93.2 | 88.67 |
| would | Would the label for this movie review be positive | 88.8 | 94.33 | 88.0 | 65.8 | 93.2 | 93.0 | 89.4 | 57.6 | 92.2 | 89.67 |
| would | Would you like the movie based on this movie review | 85.4 | 93.0 | 87.2 | 61.8 | 93.6 | 93.4 | 82.2 | 75.0 | 92.2 | 82.0 |
| would | Would this movie review make people think this is a good movie | 71.2 | 93.0 | 72.2 | 68.8 | 92.8 | 92.0 | 27.4 | 80.8 | 92.8 | 85.67 |
| would | Would this movie review make people want to watch this movie | 90.2 | 89.33 | 86.0 | 66.6 | 93.4 | 92.6 | 61.4 | 82.8 | 92.2 | 86.33 |
| would | Would this movie review be positive | 82.6 | 91.33 | 87.4 | 60.4 | 93.6 | 92.8 | 88.0 | 62.0 | 92.2 | 84.67 |
| would | Would you want to watch this movie based on this movie review | **92.0** | 94.0 | 88.8 | 57.8 | 92.8 | 93.0 | 84.4 | 77.2 | 92.0 | 84.67 |

Table 9: Detailed list of prompts for Sentiment Classification with accuracy per prompt on SST and IMDB across all models. Part 1 of 2.

| property | prompt | LLaMA 30b | | OPT 1.3b | | OPT-IML 1.3b | | OPT 30b | | OPT-IML 30b | |
|---|---|---|---|---|---|---|---|---|---|---|---|
| | | SST | IMDB | SST | IMDB | SST | IMDB | SST | IMDB | SST | IMDB |
| appr. | Does this movie appr. make people think this is a good movie | 72.8 | 93.67 | 79.2 | 61.6 | 93.4 | 89.2 | 30.6 | 80.0 | 93.6 | 94.0 |
| appr. | Does this movie appr. make people want to watch this movie | 89.0 | 91.67 | 85.0 | **65.6** | 93.4 | 91.4 | 69.6 | 80.8 | 93.6 | 92.0 |
| appr. | Is this movie appr. positive | 88.2 | 91.67 | 88.8 | 61.0 | 93.4 | 90.4 | 85.0 | 67.4 | 92.8 | 93.6 |
| appr. | Do you find this movie appr. positive | 85.2 | 93.67 | 90.4 | 55.2 | 93.6 | 91.2 | **90.6** | 62.0 | 92.6 | 92.0 |
| appr. | Is the label for this movie appr. positive | 79.2 | 91.0 | 90.0 | 62.6 | 93.2 | 91.8 | 80.2 | 69.8 | 93.8 | 94.0 |
| appr. | Do you like the movie based on this movie appr. | 79.0 | 94.0 | 89.6 | 60.0 | 92.8 | 88.2 | 89.0 | 75.0 | 92.4 | 92.0 |
| appr. | Do you want to watch this movie based on this movie appr. | 78.0 | 95.33 | 89.4 | 57.2 | 92.4 | 91.0 | 88.8 | 77.4 | 93.8 | 92.4 |
| comm. | Does this movie comm. make people think this is a good movie | 65.8 | 93.33 | 78.6 | 56.4 | 92.2 | 84.2 | 23.2 | 77.8 | 90.8 | **95.6** |
| comm. | Does this movie comm. make people want to watch this movie | 88.6 | 91.33 | 85.2 | 59.4 | 92.8 | 88.0 | 39.4 | 83.2 | 93.6 | 92.0 |
| comm. | Is this movie comm. positive | 88.4 | 90.67 | 86.2 | 61.2 | 93.4 | 90.4 | 70.2 | 67.2 | 94.0 | 93.8 |
| comm. | Do you find this movie comm. positive | 87.6 | **95.67** | 89.2 | 57.0 | 93.4 | 91.2 | 86.6 | 58.6 | 93.4 | 92.0 |
| comm. | Is the label for this movie comm. positive | 85.4 | 92.67 | 89.2 | 61.2 | 93.2 | 87.6 | 71.2 | 64.0 | 93.8 | 93.8 |
| comm. | Do you like the movie based on this movie comm. | 77.8 | 94.0 | 88.0 | 64.4 | 92.8 | 86.4 | 81.8 | 73.6 | 93.4 | 91.6 |
| comm. | Do you want to watch this movie based on this movie comm. | 77.6 | 93.0 | 88.2 | 62.0 | 92.6 | 88.4 | 78.0 | 78.6 | 91.8 | 92.8 |
| critique | Does this movie critique make people think this is a good movie | 68.6 | 93.0 | 72.8 | 57.8 | 91.2 | 91.2 | 23.6 | 80.2 | 92.6 | 95.0 |
| critique | Does this movie critique make people want to watch this movie | 87.4 | 90.0 | 83.4 | 60.2 | 92.0 | 91.6 | 47.0 | **83.8** | 92.2 | 91.2 |
| critique | Is this movie critique positive | 88.8 | 90.0 | 87.8 | 60.4 | 93.4 | 91.2 | 76.6 | 66.4 | 92.6 | 92.8 |
| critique | Do you find this movie critique positive | 85.6 | 93.67 | 90.6 | 53.0 | 93.6 | 91.8 | 89.6 | 56.2 | 92.0 | 90.2 |
| critique | Is the label for this movie critique positive | 87.2 | 91.67 | 88.0 | 64.0 | 92.8 | 93.2 | 69.8 | 64.0 | 92.8 | 93.6 |
| critique | Do you like the movie based on this movie critique | 86.6 | 94.0 | 89.4 | 60.2 | 93.0 | 88.6 | 85.0 | 73.2 | 92.6 | 90.2 |
| critique | Do you want to watch this movie based on this movie critique | 86.6 | 93.67 | 87.6 | 57.0 | 91.2 | 90.2 | 88.8 | 76.4 | 92.2 | 89.6 |
| evaluation | Does this movie evaluation make people think this is a good movie | 80.6 | 93.0 | 88.6 | 58.2 | 92.4 | 89.2 | 43.8 | 78.4 | 93.4 | 93.6 |
| evaluation | Does this movie evaluation make people want to watch this movie | 86.6 | 88.67 | 90.2 | 51.8 | 93.4 | 91.0 | 72.6 | 80.0 | 92.8 | 92.0 |
| evaluation | Is this movie evaluation positive | 89.0 | 90.0 | **91.2** | 53.0 | 93.8 | 94.4 | 82.4 | 66.8 | 92.2 | 93.0 |
| evaluation | Do you find this movie evaluation positive | 86.4 | 92.67 | 90.2 | 47.6 | **94.0** | 93.2 | 88.4 | 53.0 | 91.8 | 89.2 |
| evaluation | Is the label for this movie evaluation positive | 80.6 | 90.67 | 90.8 | 53.6 | 93.8 | 94.4 | 83.4 | 62.4 | 92.2 | 92.2 |
| evaluation | Do you like the movie based on this movie evaluation | 83.4 | 94.33 | 89.0 | 52.6 | 93.0 | 88.0 | 88.0 | 74.2 | 92.4 | 89.0 |
| evaluation | Do you want to watch this movie based on this movie evaluation | 78.8 | 94.67 | 89.2 | 49.6 | 92.6 | 92.0 | 89.2 | 76.8 | 92.6 | 89.8 |
| review | Do you find this movie review positive | 90.2 | 93.0 | 89.8 | 49.0 | 93.6 | 94.2 | 89.2 | 55.8 | 91.2 | 84.4 |
| review | Is the label for this movie review positive | 85.8 | 91.67 | 90.4 | 59.2 | 92.8 | **94.8** | 85.6 | 61.0 | 92.6 | 92.8 |
| review | Do you like the movie based on this movie review | 80.4 | 95.33 | 89.2 | 59.2 | 93.0 | 90.4 | 88.2 | 72.6 | 92.2 | 87.4 |
| review | Does this movie review make people think this is a good movie | 66.4 | 94.0 | 79.2 | 62.2 | 92.8 | 93.6 | 45.4 | 78.2 | 92.2 | 90.0 |
| review | Does this movie review make people want to watch this movie | **91.0** | 90.33 | 88.4 | 59.8 | 92.8 | 92.8 | 80.0 | 78.2 | 92.4 | 89.4 |
| review | Is this movie review positive | 89.4 | 90.67 | 90.2 | 53.4 | 93.8 | 94.0 | 89.0 | 61.0 | 92.2 | 91.4 |
| review | Do you want to watch this movie based on this movie review | 77.6 | 95.67 | 88.4 | 55.8 | 92.0 | 92.8 | 88.0 | 76.0 | 92.4 | 86.2 |

Table 10: Detailed list of prompts for Sentiment Classification with accuracy per prompt on SST and IMDB across all models. Part 2 of 2.

| prop. | prompt | LLaMA 30b | | OPT 1.3b | | OPT-IML 1.3b | | OPT 30b | | OPT-IML 30b | |
|---|---|---|---|---|---|---|---|---|---|---|---|
| | | BoolQ | ARC-E | BoolQ | ARC-E | BoolQ | ARC-E | BoolQ | ARC-E | BoolQ | ARC-E |
| | random | 50 | 25 | 50 | 25 | 50 | 25 | 50 | 25 | 50 | 25 |
| - | <null prompt> | 64.0 | 75.0 | 61.5 | 26.13 | **68.0** | 29.29 | 68.0 | 28.28 | 72.0 | 63.64 |
| ind. | You ans. the q. | 81.0 | 79.17 | 62.0 | 30.65 | 62.5 | 32.16 | 66.0 | 30.3 | **73.0** | 65.66 |
| ind. | You choose the best ans. to the q. | 60.0 | 66.67 | 62.0 | 29.15 | 62.5 | 32.66 | 60.0 | 31.31 | 72.0 | 64.65 |
| ind. | You choose this ans. | 64.0 | 69.79 | 62.0 | 27.14 | 62.0 | 33.67 | 43.0 | 22.22 | 69.0 | 66.67 |
| ind. | This ans. is correct | 77.0 | 75.0 | 62.5 | 29.15 | 63.0 | 31.16 | 57.0 | 27.27 | 70.0 | 67.68 |
| ind. | This is the correct ans. to the q. | 81.0 | 76.04 | 62.5 | 32.16 | 63.5 | 32.66 | 61.0 | 33.33 | 73.0 | 64.65 |
| ind. | You give me the correct ans. | 77.0 | 79.17 | 62.0 | 22.61 | 64.0 | 29.65 | 64.0 | 35.35 | 69.0 | 68.69 |
| ind. | You infer the correct ans. | 82.0 | 73.96 | 62.0 | 25.13 | 64.0 | 30.15 | 64.0 | 33.33 | 70.0 | 66.67 |
| ind. | You pick the correct ans. | 66.0 | 75.0 | 62.0 | 26.63 | 63.5 | 30.65 | 47.0 | 30.3 | 70.0 | 69.7 |
| ind. | You select the most suitable ans. | 68.0 | 70.83 | 62.0 | 29.15 | 62.0 | 33.67 | 47.0 | 33.33 | 68.0 | 68.69 |
| ind. | You solve the q. by choosing the correct ans. | 72.0 | 76.04 | 61.5 | 26.63 | 63.0 | **35.68** | 47.0 | 39.39 | 72.0 | **72.73** |
| ind. | You tell me which ans. is correct | 63.0 | 63.54 | 62.0 | 25.13 | 64.5 | 29.15 | 61.0 | 34.34 | 71.0 | 65.66 |
| ind. | You think this is the correct ans. | 73.0 | 72.92 | 62.5 | 27.64 | 62.0 | 31.16 | 57.0 | 27.27 | 71.0 | 65.66 |
| inter. | Could you ans. the q. | 71.0 | 76.04 | 62.5 | 31.16 | 64.5 | 34.17 | 68.0 | 30.3 | 69.0 | 64.65 |
| inter. | Could you choose the best ans. to the q. | 59.0 | 79.17 | 62.5 | 29.15 | 63.0 | 32.16 | 68.0 | 37.37 | 68.0 | 64.65 |
| inter. | Which ans. do you choose | 66.0 | 78.12 | 62.5 | 28.64 | 64.0 | 29.65 | 62.0 | 38.38 | 65.0 | 63.64 |
| inter. | Which ans. is correct | 61.0 | 73.96 | 61.5 | 25.13 | 65.5 | 27.64 | 60.0 | 32.32 | 70.0 | 65.66 |
| inter. | Which is the correct ans. to the q. | 76.0 | 73.96 | 61.0 | 28.64 | 64.5 | 32.16 | 67.0 | 31.31 | 69.0 | 64.65 |
| inter. | Could you give me the correct ans. | 70.0 | 71.88 | 60.5 | 28.64 | 64.5 | 30.65 | 69.0 | **40.4** | 67.0 | 64.65 |
| inter. | Could you infer the correct ans. | **84.0** | 77.08 | 61.5 | 28.14 | 65.0 | 31.66 | 70.0 | 37.37 | 69.0 | 64.65 |
| inter. | Could you pick the correct ans. | 69.0 | 76.04 | 62.5 | 28.14 | 64.5 | 31.16 | 66.0 | 40.4 | 67.0 | 65.66 |
| inter. | Could you select the most suitable ans. | 63.0 | 76.04 | 62.5 | 31.66 | 63.0 | 34.17 | 67.0 | 38.38 | 67.0 | 64.65 |
| inter. | Could you solve the q. by choosing the correct ans. | 70.0 | 76.04 | 62.5 | 28.14 | 64.0 | 33.17 | 69.0 | 37.37 | 70.0 | 64.65 |
| inter. | Could you tell me which ans. is correct | 64.0 | 72.92 | 61.0 | 29.15 | 64.0 | 30.65 | 68.0 | 33.33 | 69.0 | 66.67 |
| inter. | What do you think is the correct ans. | 60.0 | 73.96 | 62.5 | 28.64 | 64.5 | 31.16 | 62.0 | 36.36 | 70.0 | 66.67 |
| imp. | ans. the q. | 65.0 | 79.17 | 62.5 | **32.66** | 63.5 | 30.15 | 66.0 | 30.3 | 73.0 | 67.68 |
| imp. | Choose the best ans. to the q. | 65.0 | 70.83 | 62.0 | 28.14 | 63.0 | 32.66 | 68.0 | 32.32 | 72.0 | 65.66 |
| imp. | Tell me which ans. you choose | 62.0 | 73.96 | 62.5 | 26.13 | 65.0 | 30.15 | 67.0 | 37.37 | 69.0 | 65.66 |
| imp. | Tell me which ans. is correct | 62.0 | 75.0 | 62.5 | 26.63 | 64.5 | 29.65 | 71.0 | 34.34 | 71.0 | 67.68 |
| imp. | Tell me which is the correct ans. to the q. | 71.0 | 71.88 | 62.0 | 27.64 | 64.5 | 31.16 | 68.0 | 36.36 | 71.0 | 67.68 |
| imp. | Give me the correct ans. | 78.0 | 73.96 | 61.0 | 26.63 | 64.5 | 33.17 | 68.0 | 34.34 | 69.0 | 69.7 |
| imp. | Infer the correct ans. | 78.0 | 76.04 | 62.5 | 27.64 | 66.5 | 31.66 | **73.0** | 33.33 | 72.0 | 68.69 |
| imp. | Pick the correct ans. | 61.0 | 77.08 | 62.0 | 28.14 | 64.0 | 31.16 | 58.0 | 32.32 | 71.0 | 65.66 |
| imp. | Select the most suitable ans. | 62.0 | 75.0 | 62.0 | 28.14 | 63.0 | 35.68 | 70.0 | 26.26 | 69.0 | 67.68 |
| imp. | Solve the q. by choosing the correct ans. | 50.0 | 72.92 | 62.0 | 27.64 | 64.5 | 34.67 | 71.0 | 35.35 | 71.0 | 65.66 |
| imp. | Tell me what you think is the correct ans. | 59.0 | 75.0 | 62.0 | 28.64 | 64.5 | 33.17 | 70.0 | 36.36 | 71.0 | 65.66 |
| act. | Could you ans. the q. | 71.0 | 74.87 | 62.5 | 31.16 | 64.5 | 34.17 | 64.5 | 33.67 | 62.0 | 63.82 |
| act. | Could you choose the best ans. to the q. | 59.0 | 76.96 | 62.5 | 29.15 | 63.0 | 32.16 | 61.5 | 37.19 | 61.5 | 66.83 |
| act. | Which ans. do you choose | 66.0 | 78.01 | 62.5 | 28.64 | 64.0 | 29.65 | 58.5 | 37.69 | 63.0 | 66.83 |
| act. | Could you give me the correct ans. | 70.0 | 74.35 | 60.5 | 28.64 | 64.5 | 30.65 | 62.5 | 38.19 | 62.0 | 65.33 |
| act. | Could you infer the correct ans. | 84.0 | 76.44 | 61.5 | 28.14 | 65.0 | 31.66 | 64.0 | 36.18 | 62.0 | 66.33 |
| act. | Could you pick the correct ans. | 69.0 | 77.49 | 62.5 | 28.14 | 64.5 | 31.16 | 61.0 | 38.19 | 62.0 | 67.34 |
| act. | Could you select the most suitable ans. | 63.0 | 76.44 | 62.5 | 31.66 | 63.0 | 34.17 | 62.0 | 38.19 | 62.0 | 67.84 |
| act. | Could you solve the q. by choosing the correct ans. | 70.0 | 75.39 | 62.5 | 28.14 | 64.0 | 33.17 | 64.5 | 37.69 | 63.5 | 66.33 |
| act. | Could you tell me which ans. is correct | 64.0 | 71.73 | 61.0 | 29.15 | 64.0 | 30.65 | 60.0 | 35.68 | 62.0 | 68.84 |
| act. | What do you think is the correct ans. | 60.0 | 75.92 | 62.5 | 28.64 | 64.5 | 31.16 | 57.5 | 34.67 | 63.0 | 67.34 |
| pass. | Could the q. be ans.ed | 74.0 | 70.68 | 63.0 | 31.16 | 65.0 | 27.64 | 63.0 | 32.16 | 63.5 | 60.3 |
| pass. | Could the best ans. to the q. be chosen | 74.0 | 74.35 | 62.5 | 27.14 | 62.5 | 30.65 | 61.0 | 28.14 | 63.5 | 63.82 |
| pass. | Which ans. could be chosen | 71.0 | 74.35 | 62.5 | 26.63 | 64.5 | 30.65 | 61.5 | 33.67 | 65.0 | 66.83 |
| pass. | Could the correct ans. be given | 74.0 | 72.25 | 62.5 | 26.63 | 63.0 | 31.66 | 61.5 | 32.16 | 63.0 | 62.31 |
| pass. | Could the correct ans. be inferred | 77.0 | 66.49 | 63.0 | 28.14 | 64.5 | 27.14 | 64.5 | 32.16 | 61.5 | 62.31 |
| pass. | Could the correct ans. be picked | 71.0 | 75.92 | 63.0 | 23.62 | 63.5 | 29.15 | 60.0 | 29.65 | 64.0 | 63.32 |
| pass. | Could the most suitable ans. be selected | 75.0 | 70.68 | 62.5 | 27.64 | 62.5 | 32.16 | 61.5 | 33.17 | 63.5 | 65.83 |
| pass. | Could the q. be solved by choosing the correct ans. | 75.0 | 71.73 | **63.5** | 30.15 | 63.0 | 30.65 | 60.0 | 34.67 | 63.0 | 62.81 |
| pass. | Could it be told which ans. is correct | 72.0 | 75.39 | 61.0 | 29.65 | 62.5 | 29.65 | 60.5 | 31.66 | 63.5 | 66.33 |
| pass. | What is thought to be the correct ans. | 76.0 | 78.01 | 62.0 | 30.65 | 64.5 | 31.16 | 62.0 | 36.68 | 62.5 | 67.84 |
| can | Which ans. can you choose | 61.0 | 75.92 | 63.0 | 29.15 | 64.0 | 30.65 | 60.5 | 35.68 | 65.5 | 66.83 |
| can | Which ans. can be correct | 59.5 | 72.25 | 61.5 | 27.14 | 64.0 | 29.15 | 62.5 | 33.67 | 63.5 | 65.33 |
| can | Which can be the correct ans. to the q. | 66.5 | 69.63 | 60.5 | 27.14 | 64.0 | 32.16 | 64.0 | 33.67 | 62.5 | 67.34 |
| can | What do you think can be the correct ans. | 71.0 | 77.49 | 62.0 | 28.64 | 62.5 | 33.67 | 61.5 | 34.67 | 63.5 | 65.33 |
| could | Which ans. could you choose | 64.0 | 75.92 | 63.0 | 30.15 | 64.5 | 31.16 | 62.0 | 35.18 | 64.0 | 66.83 |
| could | Which ans. could be correct | 57.0 | 72.25 | 61.5 | 27.64 | 64.0 | 28.64 | 60.5 | 34.67 | 63.5 | 66.33 |
| could | Which could be the correct ans. to the q. | 64.0 | 69.63 | 61.0 | 27.64 | 63.0 | 32.66 | 63.0 | 32.66 | 62.5 | 67.84 |
| could | What do you think could be the correct ans. | 70.0 | 77.49 | 62.0 | 27.64 | 61.5 | 32.66 | 60.0 | 33.17 | 64.0 | 64.82 |
| may | Which ans. may you choose | 62.5 | 78.01 | 62.5 | 28.14 | 64.0 | 32.66 | 58.0 | 34.17 | 63.5 | 66.33 |
| may | Which ans. may be correct | 57.5 | 73.3 | 62.0 | 29.15 | 65.0 | 30.15 | 61.0 | 31.66 | 64.0 | 66.83 |
| may | Which may be the correct ans. to the q. | 64.0 | 69.11 | 61.5 | 26.63 | 64.0 | 32.16 | 62.0 | 29.15 | 63.0 | 67.34 |
| may | What do you think may be the correct ans. | 66.5 | 76.44 | 62.5 | 29.15 | 63.0 | 32.16 | 61.0 | 34.67 | 63.5 | 65.83 |
| might | Which ans. might you choose | 63.0 | 75.39 | 63.0 | 28.64 | 65.0 | 32.16 | 61.5 | 33.17 | 62.0 | 66.83 |
| might | Which ans. might be correct | 55.5 | 74.35 | 61.5 | 28.64 | 64.0 | 29.65 | 61.0 | 33.67 | 63.0 | 65.83 |
| might | Which might be the correct ans. to the q. | 65.0 | 72.77 | 61.0 | 27.64 | 63.5 | 31.16 | 61.0 | 31.66 | 62.0 | 66.83 |
| might | What do you think might be the correct ans. | 70.5 | **79.58** | 62.0 | 28.64 | 63.0 | 32.66 | 58.0 | 32.66 | 63.0 | 64.82 |
| must | Which ans. must you choose | 63.0 | 76.96 | 63.0 | 27.64 | 64.0 | 30.65 | 57.0 | 23.62 | 65.5 | 67.84 |
| must | Which ans. must be correct | 56.5 | 76.44 | 62.0 | 27.64 | 64.0 | 28.64 | 54.5 | 28.64 | 66.0 | 65.83 |
| must | Which must be the correct ans. to the q. | 70.5 | 72.77 | 60.5 | 28.14 | 63.5 | 32.16 | 60.5 | 29.15 | 66.0 | 67.84 |
| must | What do you think must be the correct ans. | 70.0 | 76.96 | 62.5 | 29.65 | 62.5 | 30.65 | 57.5 | 35.68 | 64.0 | 66.33 |
| should | Which ans. should you choose | 69.0 | 75.92 | 62.0 | 31.16 | 64.5 | 30.65 | 56.0 | 23.12 | 64.0 | 68.34 |
| should | Which ans. should be correct | 65.5 | 75.92 | 61.5 | 28.64 | 64.5 | 29.15 | 57.0 | 30.15 | 64.5 | 67.84 |
| should | Which should be the correct ans. to the q. | 69.0 | 74.35 | 61.5 | 27.14 | 64.0 | 30.65 | 63.5 | 30.15 | 62.5 | 67.84 |
| should | What do you think should be the correct ans. | 67.5 | 76.44 | 62.5 | 28.14 | 63.5 | 31.16 | 59.0 | 35.68 | 64.5 | 65.33 |
| would | Which ans. would you choose | 62.5 | 78.01 | 62.0 | 28.14 | 64.0 | 30.15 | 59.5 | 35.18 | 65.0 | 66.83 |
| would | Which ans. would be correct | 64.0 | 77.49 | 61.0 | 29.15 | 64.0 | 28.14 | 60.5 | 34.17 | 64.0 | 66.83 |
| would | Which would be the correct ans. to the q. | 68.5 | 75.92 | 60.5 | 27.14 | 64.5 | 31.16 | 63.0 | 34.17 | 62.5 | 68.34 |
| would | What do you think would be the correct ans. | 73.5 | 76.96 | 62.5 | 29.15 | 63.5 | 30.65 | 59.5 | 35.18 | 64.5 | 66.33 |

Table 11: Detailed list of prompts for Question Answering with accuracy per prompt on BoolQ and ARC-E across all models. Part 1 of 2.

| prop. | prompt | LLaMA 30b | | OPT 1.3b | | OPT-IML 1.3b | | OPT 30b | | OPT-IML 30b | |
|---|---|---|---|---|---|---|---|---|---|---|---|
| | | BoolQ | ARC-E | BoolQ | ARC-E | BoolQ | ARC-E | BoolQ | ARC-E | BoolQ | ARC-E |
| pro. | Could you choose the pro. ans. to the q. | 62.0 | 77.15 | 63.4 | 27.16 | 64.0 | 34.81 | 62.6 | 31.39 | 65.2 | 67.61 |
| pro. | Which ans. is pro. | 67.0 | 75.89 | 62.4 | 25.75 | 64.2 | 29.38 | 58.8 | 30.38 | 64.4 | 68.81 |
| pro. | Which is the pro. ans. to the q. | 70.2 | 73.58 | 61.8 | 26.36 | 63.2 | 33.8 | 60.0 | 30.99 | 65.0 | 70.02 |
| pro. | Could you give me the pro. ans. | 68.2 | 73.79 | 62.4 | 28.97 | 64.6 | 33.8 | 63.2 | 32.19 | 62.0 | 68.61 |
| pro. | Could you infer the pro. ans. | 76.8 | 76.94 | 61.6 | 26.96 | 64.0 | 33.8 | 61.8 | 31.79 | 62.8 | 68.01 |
| pro. | Could you pick the pro. ans. | 63.2 | 79.04 | 63.4 | 26.96 | 63.4 | 33.8 | 62.2 | 31.59 | 63.6 | 68.81 |
| pro. | Could you select the pro. ans. | 61.2 | **79.87** | 63.4 | **29.18** | 63.6 | 34.81 | 62.6 | 34.61 | 63.8 | 68.61 |
| pro. | Could you solve the q. by choosing the pro. ans. | 63.6 | 76.31 | 63.4 | 28.37 | 63.8 | 35.21 | 61.2 | 32.6 | 64.6 | 67.61 |
| pro. | Could you tell me which ans. is pro. | 64.6 | 68.76 | 62.2 | 25.96 | 63.4 | 31.39 | 61.8 | 33.2 | 63.2 | **70.22** |
| pro. | What do you think is the pro. ans. | 63.4 | 77.99 | 63.2 | 27.57 | 63.4 | 34.81 | 62.6 | 32.6 | 64.4 | 68.41 |
| right | Could you choose the right ans. to the q. | 58.6 | 79.04 | 63.2 | 27.36 | 64.2 | 33.6 | 63.8 | 31.39 | 65.0 | 67.4 |
| right | Which ans. is right | 50.4 | 72.75 | 62.0 | 26.56 | 64.6 | 31.79 | 58.0 | 31.59 | 66.4 | 68.61 |
| right | Which is the right ans. to the q. | 71.0 | 74.21 | 61.0 | 26.56 | 63.4 | 33.6 | 59.6 | 31.19 | 65.0 | 69.01 |
| right | Could you give me the right ans. | 69.2 | 74.84 | 62.8 | 27.36 | 64.6 | 35.01 | 63.2 | 31.79 | 64.4 | 68.41 |
| right | Could you infer the right ans. | **79.0** | 77.57 | 62.0 | 27.36 | 64.2 | 34.0 | 62.2 | 29.38 | 62.8 | 66.8 |
| right | Could you pick the right ans. | 62.8 | 78.62 | 63.4 | 26.36 | 63.6 | 34.81 | 62.2 | 29.78 | 64.2 | 68.21 |
| right | Could you select the right ans. | 56.0 | 79.04 | 63.4 | 27.36 | 63.6 | 34.21 | 62.6 | **35.01** | 63.8 | 68.21 |
| right | Could you solve the q. by choosing the right ans. | 64.0 | 77.15 | 63.4 | 27.57 | 64.4 | 35.21 | 61.0 | 32.39 | 64.4 | 66.6 |
| right | Could you tell me which ans. is right | 59.0 | 69.39 | 61.8 | 27.16 | 63.6 | 33.6 | 60.4 | 33.4 | 63.6 | 69.82 |
| right | What do you think is the right ans. | 57.6 | 78.41 | 63.4 | 27.36 | 63.2 | 35.01 | 58.2 | 31.79 | 64.8 | 68.01 |
| corr. | Could you choose the corr. ans. to the q. | 61.2 | 78.2 | 63.4 | 27.57 | 63.6 | 34.81 | 63.8 | 31.79 | 64.6 | 67.61 |
| corr. | Which ans. is corr. | 56.0 | 75.68 | 61.8 | 25.75 | 64.4 | 29.18 | 55.0 | 31.19 | 65.0 | 68.41 |
| corr. | Which is the corr. ans. to the q. | 69.0 | 73.79 | 60.4 | 27.97 | 63.6 | 33.4 | 59.6 | 29.78 | 64.4 | 69.22 |
| corr. | Could you give me the corr. ans. | 67.8 | 74.84 | 61.4 | 28.37 | 64.4 | 34.21 | 62.4 | 32.6 | 64.2 | 67.2 |
| corr. | Could you infer the corr. ans. | 77.6 | 77.78 | 61.8 | 28.57 | 64.2 | 33.6 | 62.6 | 30.58 | 63.0 | 67.2 |
| corr. | Could you pick the corr. ans. | 59.4 | 78.62 | 63.2 | 27.36 | 63.4 | 33.8 | 60.8 | 30.78 | 64.0 | 68.81 |
| corr. | Could you select the corr. ans. | 60.0 | 78.41 | **63.6** | 28.57 | 63.2 | 35.01 | 60.0 | 33.6 | 64.4 | 68.01 |
| corr. | Could you solve the q. by choosing the corr. ans. | 61.6 | 77.15 | 62.6 | 27.77 | 63.8 | 35.61 | 62.0 | 32.19 | 64.4 | 67.61 |
| corr. | Could you tell me which ans. is corr. | 61.4 | 69.6 | 61.6 | 27.16 | 63.4 | 33.2 | 59.2 | 32.8 | 63.6 | 69.22 |
| corr. | What do you think is the corr. ans. | 55.0 | 77.57 | 63.0 | 27.16 | 63.8 | 35.41 | 57.6 | 31.99 | 64.8 | 68.81 |
| appr. | Could you choose the appr. ans. to the q. | 63.0 | 77.99 | 63.4 | 27.16 | 64.4 | 33.6 | 63.4 | 30.38 | 65.0 | 68.41 |
| appr. | Which ans. is appr. | 69.4 | 75.68 | 62.4 | 27.16 | 63.6 | 29.98 | 58.2 | 32.19 | 66.6 | 68.61 |
| appr. | Which is the appr. ans. to the q. | 70.6 | 73.17 | 61.4 | 26.56 | 63.2 | 33.0 | 59.2 | 31.59 | 65.2 | 69.82 |
| appr. | Could you give me the appr. ans. | 69.4 | 74.21 | 62.2 | 27.57 | 64.2 | 34.61 | 63.2 | 33.0 | 63.2 | 69.01 |
| appr. | Could you infer the appr. ans. | 75.2 | 76.52 | 61.8 | 27.97 | 64.6 | 32.6 | 62.4 | 30.78 | 62.8 | 68.41 |
| appr. | Could you pick the appr. ans. | 64.4 | 77.99 | 63.4 | 27.57 | 63.8 | 34.0 | 61.8 | 32.19 | 64.4 | 69.62 |
| appr. | Could you select the appr. ans. | 62.4 | 78.41 | 63.4 | 28.97 | 64.0 | 35.01 | 62.8 | 34.21 | 64.2 | 68.81 |
| appr. | Could you solve the q. by choosing the appr. ans. | 62.8 | 75.89 | 63.4 | 28.57 | 64.4 | 34.21 | 61.8 | 31.99 | 64.6 | 67.61 |
| appr. | Could you tell me which ans. is appr. | 67.6 | 71.28 | 62.0 | 27.97 | 63.8 | 31.99 | 60.8 | 33.2 | 64.0 | 70.02 |
| appr. | What do you think is the appr. ans. | 67.6 | 77.36 | 63.0 | 27.77 | 63.8 | 34.41 | 59.6 | 33.6 | 65.0 | 68.81 |
| ans. | Could you choose the best ans. to the q. | 55.6 | 76.73 | 63.4 | 28.57 | 63.6 | 33.6 | 63.0 | 32.6 | 63.4 | 68.21 |
| ans. | Which ans. do you choose | 63.0 | 78.83 | 63.0 | 26.76 | 63.8 | 33.6 | 57.2 | 33.6 | 65.2 | 68.61 |
| ans. | Which ans. is corr. | 56.0 | 75.68 | 61.8 | 25.75 | 64.4 | 29.18 | 55.0 | 31.19 | 65.0 | 68.41 |
| ans. | Which is the corr. ans. to the q. | 69.0 | 73.79 | 60.4 | 27.97 | 63.6 | 33.4 | 59.6 | 29.78 | 64.4 | 69.22 |
| ans. | Could you give me the corr. ans. | 67.8 | 74.84 | 61.4 | 28.37 | 64.4 | 34.21 | 62.4 | 32.6 | 64.2 | 67.2 |
| ans. | Could you infer the corr. ans. | 77.6 | 77.78 | 61.8 | 28.57 | 64.2 | 33.6 | 62.6 | 30.58 | 63.0 | 67.2 |
| ans. | Could you pick the corr. ans. | 59.4 | 78.62 | 63.2 | 27.36 | 63.4 | 33.8 | 60.8 | 30.78 | 64.0 | 68.81 |
| ans. | Could you select the most suitable ans. | 53.6 | 77.36 | 63.4 | 29.18 | 63.6 | 35.01 | 61.4 | 33.2 | 63.6 | 69.82 |
| ans. | Could you solve the q. by choosing the corr. ans. | 61.6 | 77.15 | 62.6 | 27.77 | 63.8 | 35.61 | 62.0 | 32.19 | 64.4 | 67.61 |
| ans. | Could you tell me which ans. is corr. 61.4 | 69.6 | 61.6 | 27.36 | 63.4 | 33.2 | 59.2 | 32.8 | 63.6 | 69.22 | |
| ans. | What do you think is the corr. ans. | 55.0 | 77.57 | 63.0 | 27.16 | 63.8 | 35.41 | 57.6 | 31.99 | 64.8 | 68.81 |
| reply | Could you choose the best reply to the q. | 60.4 | 75.05 | 63.6 | 28.57 | 64.2 | 35.61 | 63.0 | 29.18 | 64.2 | 67.0 |
| reply | Which reply do you choose | 59.6 | 77.57 | 62.8 | 27.77 | 64.6 | 33.6 | 54.0 | 30.58 | 66.2 | 68.21 |
| reply | Which reply is corr. | 56.8 | 72.54 | 60.0 | 28.17 | 63.2 | 28.57 | 58.2 | 32.8 | 66.4 | 68.41 |
| reply | Which is the corr. reply to the q. | 68.4 | 72.33 | 61.2 | 26.56 | 63.6 | 33.0 | 59.8 | 30.58 | 65.8 | 69.01 |
| reply | Could you give me the corr. reply | 69.0 | 75.47 | 61.6 | 27.36 | 64.0 | 34.0 | 63.6 | 32.8 | 64.0 | 66.6 |
| reply | Could you infer the corr. reply | 77.2 | 77.57 | 62.4 | 26.96 | 64.0 | 33.8 | 62.2 | 29.18 | 63.6 | 66.6 |
| reply | Could you pick the corr. reply | 59.0 | 77.78 | 63.4 | 27.97 | 63.2 | 33.8 | 61.6 | 32.19 | 63.8 | 66.6 |
| reply | Could you select the most suitable reply | 56.0 | 76.94 | 63.2 | 27.36 | 64.0 | 35.21 | 62.4 | 30.99 | 63.6 | 68.81 |
| reply | Could you solve the q. by choosing the corr. reply | 62.8 | 76.94 | 63.4 | 28.97 | 64.0 | 34.81 | 61.4 | 32.8 | 64.8 | 66.8 |
| reply | Could you tell me which reply is corr. | 58.4 | 69.39 | 60.0 | 26.76 | 63.8 | 30.78 | 60.0 | 31.39 | 63.8 | 69.42 |
| reply | What do you think is the corr. reply | 64.6 | 78.2 | 62.8 | 28.57 | 64.6 | 34.41 | 61.8 | 31.59 | 64.8 | 68.41 |
| resp. | Could you choose the best resp. to the q. | 62.8 | 77.78 | 63.6 | 28.17 | 64.0 | 33.0 | 62.4 | 31.39 | 63.6 | 67.2 |
| resp. | Which resp. do you choose | 64.0 | 78.62 | 62.8 | 28.57 | 64.0 | 33.0 | 58.6 | 31.39 | 65.6 | 69.62 |
| resp. | Which resp. is corr. | 59.2 | 77.15 | 60.6 | 27.36 | 63.4 | 27.97 | 58.8 | 31.79 | 65.6 | 68.21 |
| resp. | Which is the corr. resp. to the q. | 68.4 | 74.21 | 61.2 | 26.16 | 64.0 | 32.6 | 62.0 | 30.99 | 65.8 | 70.22 |
| resp. | Could you give me the corr. resp. | 71.0 | 73.79 | 62.0 | 28.97 | 64.2 | 32.8 | 62.8 | 32.19 | 63.8 | 67.61 |
| resp. | Could you infer the corr. resp. | 77.4 | 77.57 | 63.0 | 26.96 | 64.2 | 32.8 | 62.6 | 30.38 | 62.8 | 66.8 |
| resp. | Could you pick the corr. resp. | 66.6 | 78.83 | 63.4 | 26.76 | 63.6 | 34.41 | 61.2 | 31.19 | 63.4 | 67.61 |
| resp. | Could you select the most suitable resp. | 63.4 | 76.94 | 63.2 | 27.57 | 63.6 | 34.61 | 62.0 | 30.18 | 63.8 | 69.82 |
| resp. | Could you solve the q. by choosing the corr. resp. | 67.8 | 76.31 | 63.4 | 27.16 | 64.4 | 36.02 | 63.4 | 32.6 | 64.6 | 67.2 |
| resp. | Could you tell me which resp. is corr. | 60.4 | 71.28 | 59.2 | 25.96 | 62.8 | 31.39 | 61.0 | 33.2 | 63.0 | 69.01 |
| resp. | What do you think is the corr. resp. | 64.8 | 77.78 | 63.4 | 27.16 | 64.6 | 33.8 | 61.2 | 32.8 | 65.4 | 68.01 |
| sol. | Could you choose the best sol. to the q. | 57.0 | 74.21 | 63.2 | 28.77 | 64.2 | 32.8 | 63.8 | 29.38 | 63.6 | 67.2 |
| sol. | Which sol. do you choose | 65.4 | 75.89 | 63.6 | 27.57 | 64.0 | 32.39 | 59.8 | 29.98 | 65.0 | 68.61 |
| sol. | Which sol. is corr. | 55.4 | 64.36 | 62.4 | 27.97 | 62.8 | 29.18 | 57.4 | 29.98 | 65.6 | 67.0 |
| sol. | Which is the corr. sol. to the q. | 71.4 | 71.7 | 62.2 | 25.75 | 64.2 | 33.0 | 62.8 | 31.99 | 66.0 | 69.62 |
| sol. | Could you give me the corr. sol. | 72.6 | 73.79 | 62.2 | 28.17 | 64.0 | 33.2 | 61.0 | 32.19 | 62.8 | 66.8 |
| sol. | Could you infer the corr. sol. | 74.2 | 76.94 | 62.2 | 27.36 | 64.2 | 33.6 | 60.0 | 28.97 | 63.2 | 66.6 |
| sol. | Could you pick the corr. sol. | 58.4 | 77.57 | 63.6 | 28.97 | 64.2 | 33.2 | 58.8 | 31.79 | 64.4 | 67.2 |
| sol. | Could you select the most suitable sol. | 52.2 | 74.63 | 63.4 | 27.97 | 64.0 | 33.6 | 62.4 | 29.78 | 63.6 | 68.21 |
| sol. | Could you solve the q. by choosing the corr. sol. | 62.4 | 75.26 | 63.4 | 28.97 | 64.0 | **37.02** | 58.8 | 32.19 | 64.6 | 66.0 |
| sol. | Could you tell me which sol. is corr. | 59.0 | 64.99 | 63.6 | 27.77 | 63.6 | 32.39 | 60.8 | 31.59 | 63.4 | 68.41 |
| sol. | What do you think is the corr. sol. | 62.0 | 74.84 | 63.4 | 28.97 | 63.6 | 34.21 | 61.0 | 31.99 | 65.4 | 67.4 |

Table 12: Detailed list of prompts for Question Answering with accuracy per prompt on BoolQ and ARC-E across all models. Part 1 of 2.

| property | prompt | LLaMA 30b | | OPT 1.3b | | OPT-IML 1.3b | | OPT 30b | | OPT-IML 30b | |
|---|---|---|---|---|---|---|---|---|---|---|---|
| | | RTE | CB | RTE | CB | RTE | CB | RTE | CB | RTE | CB |
| null | | 45.0 | 55.2 | 40.0 | 57.6 | 46.5 | 61.6 | 46.6 | 66.8 | 41.2 | 79.2 |
| ind. | Given "[p]" you can assume that "[h]" | 51.8 | 48.8 | 51.8 | 53.2 | 63.8 | 63.6 | 50.4 | 46.4 | 71.8 | 77.2 |
| ind. | Given "[p]" the claim "[h]" is correct | 48.6 | 47.2 | 54.4 | 52.4 | 62.6 | 63.2 | 50.4 | 47.6 | 73.8 | 81.6 |
| ind. | Given "[p]" you can deduce that "[h]" | 52.4 | 50.8 | 53.4 | 54.4 | 64.0 | 63.2 | 50.6 | 46.0 | 74.0 | 80.0 |
| ind. | Given "[p]" it follows that "[h]" | 52.0 | 47.6 | 52.4 | 50.8 | 65.0 | 62.4 | 51.6 | 46.0 | 71.2 | 79.2 |
| ind. | Given "[p]" this implies that "[h]" | 60.2 | 48.8 | 53.2 | 52.8 | 63.2 | 62.8 | 53.8 | 48.4 | 71.2 | 74.0 |
| ind. | Given "[p]" you can infer that "[h]" | 55.2 | 51.2 | 53.0 | 53.2 | 64.0 | 62.0 | 50.8 | 46.0 | 73.4 | 80.0 |
| ind. | Given "[p]" you are justified in saying that "[h]" | 50.0 | 48.0 | 52.6 | 52.4 | 65.2 | 62.4 | 51.2 | 46.4 | 76.4 | 84.0 |
| ind. | Given premise "[p]" and hypothesis "[h]" the label is ent. | 55.2 | 64.8 | 48.6 | 48.0 | 64.4 | 61.2 | 49.0 | 46.0 | 76.4 | **86.4** |
| ind. | Given "[p]" you can reason that "[h]" | 51.0 | 48.8 | **54.8** | 49.6 | 64.4 | 62.8 | 50.0 | 46.8 | 73.4 | 77.2 |
| ind. | The relationship between "[p]" and "[h]" is ent. | 49.4 | 47.2 | 52.6 | 54.8 | 62.0 | 60.0 | 48.6 | 46.4 | 67.0 | 74.8 |
| ind. | Given "[p]" it is true that "[h]" | 49.0 | 46.8 | 52.0 | 53.6 | 64.6 | 60.8 | 49.2 | 47.6 | 75.4 | 80.4 |
| ind. | "[p]" Using only the above description "[h]" is correct | 51.2 | 46.4 | 52.4 | 47.6 | 63.8 | 62.8 | 50.8 | 46.8 | 71.2 | 81.2 |
| inter. | Given "[p]" can you assume that "[h]"? | 52.6 | 56.8 | 51.0 | 60.8 | 65.0 | 67.2 | 51.8 | 56.8 | 73.0 | 74.8 |
| inter. | Given "[p]" is the claim "[h]" correct | 53.2 | 48.4 | 50.8 | 59.6 | 67.4 | 59.6 | 52.2 | 66.0 | 73.4 | 79.6 |
| inter. | Given "[p]" can you deduce that "[h]" | 52.0 | 57.6 | 53.6 | 58.0 | 64.8 | 64.0 | 51.2 | 60.4 | 73.8 | 77.2 |
| inter. | Given "[p]" does it follow that "[h]" | 49.0 | 51.6 | 50.2 | 60.0 | 66.2 | 62.8 | 56.4 | 65.2 | 68.8 | 65.6 |
| inter. | Given "[p]" does this imply that "[h]" | 53.4 | 61.6 | 52.8 | 60.8 | 66.0 | 62.4 | 52.0 | 57.6 | 70.2 | 68.4 |
| inter. | Given "[p]" can you infer that "[h]" | 56.0 | **76.0** | 49.0 | 58.8 | 65.4 | 64.8 | 52.4 | 56.8 | 72.0 | 77.2 |
| inter. | Given "[p]" are you justified in saying that "[h]" | 56.2 | 53.2 | 50.6 | 54.8 | 65.8 | 59.2 | 53.4 | 59.6 | 75.4 | 79.6 |
| inter. | Given premise "[p]" and hypothesis "[h]" is the label ent. | 53.8 | 46.4 | 51.0 | 49.6 | 62.8 | 68.0 | 49.0 | 46.4 | 75.4 | 81.2 |
| inter. | Given "[p]" can you reason that "[h]" | 59.2 | 59.6 | 50.8 | 58.0 | 64.6 | 60.8 | 56.0 | 67.2 | 75.2 | 78.0 |
| inter. | Is the relationship between "[p]" and "[h]" ent. | 49.2 | 35.2 | 48.4 | 63.6 | 63.4 | 57.6 | 49.0 | 46.0 | 67.8 | 67.2 |
| inter. | Given "[p]" is it true that "[h]" | 59.8 | 54.0 | 52.4 | 56.0 | 69.2 | 64.0 | 57.0 | 67.2 | 75.2 | 77.6 |
| inter. | "[p]" Using only the above description is "[h]" correct | 54.0 | 42.0 | 52.2 | 58.0 | 65.0 | 57.6 | 53.2 | 62.8 | 70.8 | 78.8 |
| imp. | Given "[p]" tell me if you can assume that "[h]" | 56.4 | 48.8 | 53.0 | 52.0 | 63.0 | 60.0 | 49.2 | 55.2 | 77.2 | 80.0 |
| imp. | Given "[p]" tell me if the claim "[h]" is correct | **65.8** | 56.0 | 53.6 | 49.2 | 64.4 | 58.0 | 57.0 | 69.2 | 76.8 | 85.2 |
| imp. | Given "[p]" tell me if you can deduce that "[h]" | 54.4 | 51.2 | 53.6 | 54.0 | 64.4 | 63.6 | 49.4 | 57.2 | 77.6 | 80.4 |
| imp. | Given "[p]" tell me if it follows that "[h]" | 54.6 | 49.2 | 52.6 | 51.6 | 64.8 | 64.4 | 55.4 | 60.4 | 73.4 | 78.0 |
| imp. | Given "[p]" tell me if this implies that "[h]" | 57.2 | 65.2 | 52.4 | 49.6 | 63.8 | 62.4 | 52.2 | 56.4 | 71.4 | 75.2 |
| imp. | Given "[p]" tell me if you can infer that "[h]" | 62.2 | 61.2 | 52.8 | 56.0 | 65.0 | 64.4 | 49.8 | 55.6 | 76.6 | 81.6 |
| imp. | Given "[p]" tell me if you are justified in saying that "[h]" | 60.6 | 52.0 | 53.8 | 49.2 | 64.6 | 57.2 | 53.0 | 55.6 | 78.0 | 81.2 |
| imp. | Given premise "[p]" and hypothesis "[h]" tell me if the label is ent. | 52.2 | 50.0 | 53.6 | 48.4 | 64.2 | 64.4 | 49.0 | 47.2 | 75.6 | 83.2 |
| imp. | Given "[p]" tell me if you can reason that "[h]" | 60.8 | 51.6 | 52.8 | 50.4 | 63.8 | 61.2 | 56.0 | 70.0 | 77.2 | 80.4 |
| imp. | Tell me if the relationship between "[p]" and "[h]" is ent. | 46.8 | 34.0 | 50.8 | 54.8 | 63.8 | 58.0 | 51.4 | 49.2 | 67.4 | 70.8 |
| imp. | Given "[p]" tell me if it is true that "[h]" | 63.2 | 61.6 | 52.0 | 52.8 | 66.6 | 64.8 | 54.6 | 62.4 | 76.6 | 80.8 |
| imp. | "[p]" Using only the above description tell me if "[h]" is correct | 58.2 | 48.0 | 51.6 | 57.6 | 67.6 | 52.4 | 55.0 | 72.8 | **79.2** | 84.4 |
| active | Given "[p]" can you assume that "[h]" | 52.6 | 56.8 | 51.0 | 60.8 | 65.0 | 67.2 | 51.8 | 56.8 | 73.0 | 74.8 |
| active | Given "[p]" can you conclude that "[h]" | 51.4 | 51.2 | 51.4 | 56.8 | 66.4 | 62.4 | 56.8 | 66.8 | 71.6 | 73.6 |
| active | Given "[p]" can you deduce that "[h]" | 52.0 | 57.6 | 53.6 | 58.0 | 64.8 | 64.0 | 51.2 | 60.4 | 73.8 | 77.2 |
| active | Given "[p]" does it follow that "[p]" | 49.8 | 38.8 | 53.6 | 59.6 | 59.6 | 51.2 | 50.8 | 49.2 | 63.0 | 55.6 |
| active | Given "[p]" can you guess that "[h]" | 60.2 | 65.6 | 52.8 | 56.8 | 65.8 | 64.8 | 51.0 | 72.8 | 74.8 | 74.4 |
| active | Given "[p]" does this imply that "[h]" | 53.4 | 61.6 | 52.8 | 60.8 | 66.0 | 62.4 | 52.0 | 57.6 | 70.2 | 68.4 |
| active | Given premis can you infer that "[h]" | 53.4 | 47.2 | 50.0 | 50.4 | 52.8 | 39.6 | 49.2 | 45.6 | 59.2 | 47.6 |
| active | Given "[p]" can you justifiedly say that "[h]" | 57.2 | 51.2 | 51.4 | 54.4 | 65.6 | 62.0 | 57.4 | 63.2 | 76.0 | 76.0 |
| active | Given premise "[p]" and hypothesis "[h]" can you label this as ent. | 51.2 | 47.2 | 49.4 | 47.2 | 63.0 | 64.4 | 49.0 | 46.0 | 76.0 | 82.8 |
| active | Given "[p]" can you reason that "[h]" | 59.2 | 59.6 | 50.8 | 58.0 | 64.6 | 60.8 | 56.0 | 67.2 | 75.2 | 78.0 |
| passive | Given "[p]" can it be assumed that "[h]" | 54.4 | 60.0 | 52.4 | 62.8 | 64.2 | 67.2 | 52.4 | 53.2 | 71.4 | 72.8 |
| passive | Given "[p]" can it be concluded that "[h]" | 51.2 | 52.0 | 50.6 | 59.6 | 63.0 | 64.0 | 56.6 | 65.2 | 70.2 | 71.2 |
| passive | Given "[p]" can it be deduced that "[h]" | 51.6 | 54.8 | 51.2 | 61.2 | 63.0 | 62.8 | 51.4 | 57.6 | 71.0 | 73.2 |
| passive | Given "[h]" is it followed that "[p]" | 52.6 | 38.8 | 54.6 | 57.6 | 60.0 | 50.0 | 49.4 | 49.6 | 61.4 | 54.8 |
| passive | Given "[p]" can it be guessed that "[h]" | 55.6 | 54.4 | 53.6 | 58.8 | 64.6 | 64.4 | 49.8 | 59.2 | 70.2 | 70.4 |
| passive | Given "[p]" is it implied that "[h]" | 55.0 | 60.4 | 52.4 | 61.6 | 67.8 | 60.8 | 50.8 | 60.0 | 70.8 | 71.6 |
| passive | Given premis can it be inferred that "[h]" | 51.0 | 45.6 | 49.4 | 47.6 | 52.0 | 38.8 | 49.2 | 46.8 | 60.0 | 48.0 |
| passive | Given "[p]" can it justifiedly be said that "[h]" | 55.0 | 53.6 | 51.6 | 57.6 | 64.0 | 62.0 | 55.8 | 60.8 | 73.4 | 69.6 |
| passive | Given premise "[p]" and hypothesis "[h]" can this be labelled as ent. | 55.6 | 48.0 | 50.2 | 45.6 | 63.0 | 64.4 | 49.0 | 46.0 | 75.2 | 81.2 |
| passive | Given "[p]" can it be reasoned that "[h]" | 56.0 | 54.4 | 51.8 | 57.6 | 65.2 | 62.4 | 52.6 | 57.2 | 71.0 | 72.8 |
| past | Given "[p]" did you assume that "[h]" | 52.8 | 54.0 | 53.0 | 56.4 | 63.6 | 65.2 | 49.6 | 56.8 | 68.2 | 66.8 |
| past | Given "[p]" was the claim "[h]" correct | 51.4 | 54.4 | 50.6 | 60.0 | 68.2 | 58.4 | 50.8 | 66.4 | 70.8 | 74.8 |
| past | Given "[p]" did you deduce that "[h]" | 55.2 | 58.8 | 52.8 | 59.2 | 65.8 | 61.2 | 51.4 | 66.8 | 70.4 | 71.6 |
| past | Given "[p]" did it follow that "[h]" | 49.8 | 46.4 | 51.6 | 56.8 | 65.4 | 62.4 | 59.0 | 66.0 | 67.8 | 65.6 |
| past | Given "[p]" did this imply that "[h]" | 51.0 | 52.2 | 52.2 | 61.2 | 65.0 | 64.4 | 51.6 | 58.4 | 70.0 | 67.6 |
| past | Given "[p]" did you infer that "[h]" | 52.2 | 64.4 | 51.4 | 58.4 | 65.4 | 63.2 | 52.0 | 64.4 | 69.2 | 70.8 |
| past | Given "[p]" were you justified in saying that "[h]" | 55.4 | 54.4 | 51.0 | 53.2 | 66.4 | 63.2 | 54.0 | 59.6 | 74.2 | 77.6 |
| past | Given premise "[p]" and hypothesis "[h]" was the label ent. | 52.8 | 48.0 | 53.6 | 54.4 | 63.8 | 65.2 | 49.0 | 46.4 | 74.8 | 84.8 |
| past | Given "[p]" did you reason that "[h]" | 55.4 | 56.8 | 52.6 | 58.4 | 65.0 | 64.4 | 51.6 | 64.4 | 70.0 | 68.0 |
| past | Was the relationship between "[p]" and "[h]" ent. | 52.2 | 34.4 | 50.2 | 64.0 | 63.8 | 56.0 | 49.0 | 46.0 | 67.2 | 68.8 |
| past | Given "[p]" was it true that "[h]" | 57.4 | 50.8 | 53.8 | 55.2 | **70.0** | 62.0 | 55.2 | 64.8 | 75.0 | 73.2 |
| past | "[p]" Using only the above description was "[h]" correct | 51.8 | 46.0 | 53.0 | 58.8 | 65.2 | 59.2 | 52.6 | 63.6 | 70.8 | 76.0 |
| present | Given "[p]" do you assume that "[h]" | 47.6 | 50.8 | 51.0 | 57.6 | 65.8 | 65.6 | 49.6 | 57.6 | 72.6 | 72.4 |
| present | Given "[p]" is the claim "[h]" correct | 53.2 | 48.4 | 50.8 | 59.6 | 67.4 | 59.6 | 52.2 | 66.4 | 73.4 | 79.6 |
| present | Given "[p]" do you deduce that "[h]" | 52.4 | 55.6 | 50.6 | 59.6 | 66.4 | 64.4 | 50.0 | 66.8 | 70.8 | 72.0 |
| present | Given "[p]" does it follow that "[h]" | 49.0 | 51.6 | 50.2 | 60.0 | 66.2 | 62.8 | 56.4 | 64.8 | 68.8 | 65.6 |
| present | Given "[p]" does this imply that "[h]" | 53.4 | 61.6 | 52.8 | 60.8 | 66.0 | 62.4 | 52.0 | 58.0 | 70.2 | 68.4 |
| present | Given "[p]" do you infer that "[h]" | 54.0 | 64.4 | 52.0 | 60.8 | 66.2 | 64.0 | 51.4 | 65.2 | 70.6 | 73.2 |
| present | Given "[p]" are you justified in saying that "[h]" | 56.2 | 53.2 | 50.6 | 54.8 | 65.8 | 59.2 | 53.4 | 59.6 | 75.4 | 79.6 |
| present | Given premise "[p]" and hypothesis "[h]" is the label ent. | 54.4 | 50.8 | 51.4 | 54.0 | 63.4 | 66.8 | 49.0 | 46.4 | 75.2 | 84.8 |
| present | Given "[p]" do you reason that "[h]" | 52.4 | 52.0 | 51.6 | 60.0 | 66.0 | 65.2 | 51.8 | 66.4 | 70.6 | 71.6 |
| present | Is the relationship between "[p]" and "[h]" ent. | 49.2 | 35.2 | 48.4 | 63.6 | 63.4 | 57.6 | 49.0 | 46.0 | 67.8 | 67.2 |
| present | Given "[p]" is it true that "[h]" | 59.8 | 54.0 | 52.4 | 56.0 | 69.2 | 64.0 | 57.0 | 66.8 | 75.2 | 77.6 |
| present | "[p]" Using only the above description is "[h]" correct | 54.0 | 42.0 | 52.2 | 58.0 | 65.0 | 57.6 | 53.2 | 62.4 | 70.8 | 78.8 |
| future | Given "[p]" will you assume that "[h]" | 49.6 | 48.8 | 52.0 | 59.2 | 64.8 | 66.4 | 51.0 | 55.2 | 72.2 | 74.0 |
| future | Given "[p]" will the claim "[h]" be correct | 55.8 | 48.8 | 49.8 | 49.2 | 64.0 | 59.6 | 52.6 | 72.4 | 72.8 | 78.0 |
| future | Given "[p]" will you deduce that "[h]" | 51.0 | 50.4 | 52.8 | 60.4 | 64.8 | 63.2 | 51.2 | 64.4 | 71.0 | 73.6 |
| future | Given "[p]" will it follow that "[h]" | 53.6 | 52.0 | 51.4 | 58.0 | 66.6 | 62.8 | 52.8 | 61.2 | 69.6 | 68.4 |
| future | Given "[p]" will this imply that "[h]" | 52.6 | 52.8 | 53.0 | 59.6 | 65.6 | 64.0 | 50.0 | 54.8 | 70.0 | 67.2 |
| future | Given "[p]" will you infer that "[h]" | 51.4 | 54.8 | 53.2 | 60.8 | 66.4 | 65.6 | 51.0 | 62.8 | 70.0 | 73.6 |
| future | Given "[p]" will you be justified in saying that "[h]" | 53.6 | 48.0 | 53.0 | 51.2 | 65.8 | 58.8 | 52.6 | 59.2 | 74.0 | 75.6 |
| future | Given premise "[p]" and hypothesis "[h]" will the label be ent. | 52.4 | 48.0 | 53.0 | 48.8 | 63.2 | 67.2 | 49.0 | 46.0 | 75.6 | 82.0 |
| future | Given "[p]" will you reason that "[h]" | 52.2 | 46.4 | 51.8 | 55.2 | 67.4 | 63.6 | 51.8 | 65.2 | 71.8 | 71.6 |
| future | Will the relationship between "[p]" and "[h]" be ent. | 48.4 | 40.4 | 52.0 | 62.0 | 62.4 | 57.6 | 49.0 | 47.6 | 67.6 | 67.2 |
| future | Given "[p]" will it be true that "[h]" | 55.0 | 49.6 | 50.8 | 52.4 | 66.4 | 62.0 | 50.8 | 59.2 | 74.4 | 76.8 |
| future | "[p]" Using only the above description will "[h]" be correct | 54.6 | 47.6 | 52.4 | 53.6 | 65.0 | 60.8 | 55.4 | **78.4** | 70.0 | 72.4 |

Table 13: Detailed list of prompts for Natural Language Inference with accuracy per prompt on RTE and CB across all models. Part 1 of 2.

| property | prompt | LLaMA 30b | | OPT 1.3b | | OPT-IML 1.3b | | OPT 30b | | OPT-IML 30b | |
|---|---|---|---|---|---|---|---|---|---|---|---|
| | | RTE | CB | RTE | CB | RTE | CB | RTE | CB | RTE | CB |
| can | Given "[p]" can you assume that "[h]"? | 52.6 | 56.8 | 51.0 | 60.8 | 65.0 | 67.2 | 51.8 | 56.8 | 73.0 | 74.8 |
| can | Given "[p]" can the claim "[h]" be correct? | 53.8 | 48.0 | 50.8 | 52.4 | 67.4 | 57.6 | 51.8 | 68.8 | 74.8 | 77.6 |
| can | Given "[p]" can you deduce that "[h]"? | 52.0 | 57.6 | 53.6 | 58.0 | 64.8 | 64.0 | 52.0 | 60.4 | 73.8 | 77.2 |
| can | Given "[p]" can it follow that "[h]"? | 55.6 | 50.0 | 51.4 | 57.6 | 63.8 | 63.6 | 54.2 | 60.4 | 69.8 | 70.0 |
| can | Given "[p]" can this imply that "[h]"? | 53.6 | 56.4 | 52.0 | 60.8 | 65.8 | 63.2 | 50.8 | 51.2 | 70.6 | 70.0 |
| can | Given "[p]" can you infer that "[h]"? | 56.0 | 76.0 | 49.0 | 58.8 | 65.4 | 64.8 | 52.4 | 56.8 | 72.0 | 77.6 |
| can | Given "[p]" can you be justified in saying that "[h]"? | 58.4 | 51.2 | 51.2 | 57.6 | 66.4 | 59.6 | 57.8 | 62.0 | 74.2 | 75.2 |
| can | Given premise "[p]" and hypothesis "[h]" can the label be ent.? | 52.2 | 47.6 | 52.0 | 46.8 | 63.6 | 64.0 | 49.0 | 46.4 | 75.2 | 79.6 |
| can | Given "[p]" can you reason that "[h]"? | 59.2 | 59.6 | 50.8 | 58.0 | 64.6 | 60.8 | 56.0 | 67.2 | 75.2 | 78.0 |
| can | Can the relationship between "[p]" and "[h]" be ent.? | 51.6 | 38.8 | 52.8 | 59.6 | 64.2 | 56.8 | 49.0 | 47.6 | 67.4 | 68.0 |
| can | Given "[p]" can it be true that "[h]"? | 54.6 | 47.2 | 51.8 | 55.6 | 68.4 | 64.8 | 56.2 | 54.8 | 74.4 | 74.0 |
| can | "[p]" Using only the above description can "[h]" be correct? | 54.2 | 47.6 | 53.2 | 53.2 | 66.2 | 55.6 | 56.4 | 76.4 | 71.2 | 72.4 |
| could | Given "[p]" could you assume that "[h]"? | 53.4 | 58.4 | 51.2 | 60.8 | 65.2 | 66.8 | 51.0 | 56.4 | 71.6 | 72.8 |
| could | Given "[p]" could the claim "[h]" be correct? | 53.6 | 47.2 | 49.6 | 54.8 | 66.4 | 59.2 | 53.8 | 72.0 | 74.0 | 78.8 |
| could | Given "[p]" could you deduce that "[h]"? | 52.0 | 55.6 | 50.2 | 61.6 | 64.8 | 66.0 | 50.2 | 62.4 | 73.6 | 76.0 |
| could | Given "[p]" could it follow that "[h]"? | 56.0 | 48.4 | 51.6 | 57.6 | 64.2 | 62.4 | 50.2 | 54.4 | 69.8 | 70.4 |
| could | Given "[p]" could this imply that "[h]"? | 53.4 | 50.0 | 51.4 | 60.4 | 65.4 | 65.6 | 49.6 | 50.4 | 69.6 | 68.4 |
| could | Given "[p]" could you infer that "[h]"? | 53.6 | 69.2 | 51.4 | **64.4** | 65.8 | 64.4 | 52.0 | 57.2 | 72.0 | 74.8 |
| could | Given "[p]" could you be justified in saying that "[h]"? | 59.6 | 54.0 | 51.2 | 60.8 | 66.2 | 59.2 | 53.2 | 60.8 | 73.2 | 75.6 |
| could | Given premise "[p]" and hypothesis "[h]" could the label be ent.? | 53.4 | 57.2 | 51.4 | 49.6 | 62.6 | 62.8 | 49.0 | 46.0 | 74.0 | 80.0 |
| could | Given "[p]" could you reason that "[h]"? | 57.8 | 60.8 | 50.4 | 58.8 | 65.6 | 63.6 | 54.2 | 67.2 | 73.4 | 77.2 |
| could | Could the relationship between "[p]" and "[h]" be ent.? | 50.8 | 40.4 | 54.6 | 60.8 | 64.0 | 55.6 | 49.0 | 47.2 | 67.2 | 68.4 |
| could | Given "[p]" could it be true that "[h]"? | 52.2 | 52.0 | 51.8 | 56.0 | 67.2 | 64.8 | 50.6 | 56.4 | 73.8 | 74.0 |
| could | "[p]" Using only the above description could "[h]" be correct? | 52.2 | 47.2 | 52.0 | 52.8 | 64.6 | 58.0 | 53.6 | 77.2 | 71.0 | 75.6 |
| may | Given "[p]" may you assume that "[h]"? | 52.2 | 55.6 | 52.0 | 62.4 | 64.4 | **68.4** | 51.6 | 50.0 | 72.0 | 70.8 |
| may | Given "[p]" may the claim "[h]" be correct? | 55.8 | 50.0 | 52.2 | 53.2 | 66.2 | 60.4 | 54.6 | 69.2 | 74.8 | 79.6 |
| may | Given "[p]" may you deduce that "[h]"? | 51.2 | 58.0 | 51.8 | 62.0 | 64.0 | 66.0 | 50.0 | 51.2 | 72.0 | 75.2 |
| may | Given "[p]" may it follow that "[h]"? | 54.0 | 54.4 | 51.4 | 61.6 | 64.2 | 62.8 | 54.4 | 60.0 | 70.8 | 70.4 |
| may | Given "[p]" may this imply that "[h]"? | 52.2 | 51.6 | 52.8 | 55.2 | 65.4 | 66.0 | 50.6 | 52.0 | 70.0 | 67.6 |
| may | Given "[p]" may you infer that "[h]"? | 52.4 | 44.4 | 52.4 | 62.4 | 64.8 | 67.2 | 51.4 | 55.2 | 71.0 | 73.2 |
| may | Given "[p]" may you be justified in saying that "[h]"? | 54.6 | 52.0 | 52.0 | 59.6 | 66.0 | 60.0 | 54.2 | 58.4 | 73.4 | 75.6 |
| may | Given premise "[p]" and hypothesis "[h]" may the label be ent.? | 54.6 | 50.4 | 54.0 | 52.8 | 62.6 | 64.4 | 49.0 | 46.4 | 75.2 | 81.6 |
| may | Given "[p]" may you reason that "[h]"? | 53.2 | 52.8 | 52.2 | 60.4 | 65.2 | 64.8 | 53.2 | 59.2 | 71.8 | 72.0 |
| may | May the relationship between "[p]" and "[h]" be ent.? | 50.6 | 42.4 | 54.2 | 59.2 | 62.6 | 62.4 | 49.0 | 48.8 | 67.2 | 69.6 |
| may | Given "[p]" may it be true that "[h]"? | 57.0 | 48.8 | 51.8 | 54.8 | 66.2 | 66.0 | 51.2 | 52.4 | 74.6 | 77.2 |
| may | "[p]" Using only the above description may "[h]" be correct? | 50.6 | 46.8 | 52.8 | 53.2 | 64.6 | 60.0 | 54.2 | 76.4 | 68.4 | 70.4 |
| might | Given "[p]" might you assume that "[h]"? | 56.4 | 56.4 | 52.4 | 58.0 | 64.8 | 68.4 | 52.2 | 55.6 | 70.6 | 68.8 |
| might | Given "[p]" might the claim "[h]" be correct? | 52.0 | 46.8 | 49.8 | 56.0 | 66.6 | 61.2 | 52.8 | 68.4 | 73.0 | 77.2 |
| might | Given "[p]" might you deduce that "[h]"? | 60.2 | 57.6 | 53.0 | 60.8 | 65.2 | 63.6 | 49.4 | 57.2 | 70.6 | 71.2 |
| might | Given "[p]" might it follow that "[h]"? | 56.6 | 49.2 | 52.6 | 57.6 | 64.8 | 62.0 | 51.6 | 53.6 | 70.2 | 70.8 |
| might | Given "[p]" might this imply that "[h]"? | 51.6 | 53.2 | 53.0 | 58.8 | 65.2 | 66.0 | 50.8 | 50.4 | 70.2 | 68.0 |
| might | Given "[p]" might you infer that "[h]"? | 62.2 | 66.0 | 52.8 | 62.8 | 66.6 | 65.6 | 51.6 | 55.6 | 70.8 | 70.8 |
| might | Given "[p]" might you be justified in saying that "[h]"? | 59.2 | 43.2 | 52.8 | 56.0 | 66.2 | 58.4 | 51.8 | 56.8 | 73.6 | 74.8 |
| might | Given premise "[p]" and hypothesis "[h]" might the label be ent.? | 55.4 | 58.8 | 53.6 | 52.0 | 63.8 | 66.0 | 49.0 | 46.4 | 74.2 | 79.6 |
| might | Given "[p]" might you reason that "[h]"? | 61.4 | 56.4 | 52.4 | 58.4 | 65.8 | 64.8 | 51.6 | 57.2 | 70.6 | 71.6 |
| might | Might the relationship between "[p]" and "[h]" be ent.? | 46.0 | 39.2 | 53.0 | 56.0 | 65.0 | 56.8 | 49.2 | 48.4 | 67.2 | 67.6 |
| might | Given "[p]" might it be true that "[h]"? | 49.2 | 48.4 | 52.4 | 54.4 | 67.2 | 64.8 | 52.2 | 53.2 | 74.6 | 75.2 |
| might | "[p]" Using only the above description might "[h]" be correct? | 51.6 | 47.2 | 52.0 | 53.6 | 64.2 | 60.0 | 53.0 | 74.4 | 68.6 | 72.0 |
| must | Given "[p]" must you assume that "[h]"? | 53.2 | 51.6 | 51.8 | 56.4 | 64.8 | 61.2 | 50.0 | 50.0 | 69.6 | 66.0 |
| must | Given "[p]" must the claim "[h]" be correct? | 52.0 | 47.2 | 49.0 | 50.0 | 66.4 | 57.6 | 53.2 | 68.0 | 71.4 | 76.8 |
| must | Given "[p]" must you deduce that "[h]"? | 55.8 | 56.0 | 50.2 | 59.2 | 66.0 | 64.8 | 51.4 | 59.2 | 70.6 | 70.8 |
| must | Given "[p]" must it follow that "[h]"? | 53.8 | 52.0 | 51.4 | 54.4 | 65.6 | 60.4 | **61.4** | 53.2 | 68.8 | 66.4 |
| must | Given "[p]" must this imply that "[h]"? | 53.2 | 48.0 | 52.0 | 56.4 | 66.2 | 62.0 | 52.4 | 53.2 | 69.8 | 66.8 |
| must | Given "[p]" must you infer that "[h]"? | 54.6 | 57.6 | 51.2 | 58.4 | 66.6 | 64.0 | 54.0 | 56.8 | 69.8 | 70.8 |
| must | Given "[p]" must you be justified in saying that "[h]"? | 63.0 | 49.6 | 52.0 | 54.8 | 66.2 | 59.6 | 52.4 | 55.6 | 72.4 | 73.6 |
| must | Given premise "[p]" and hypothesis "[h]" must the label be ent.? | 56.6 | 51.2 | 52.2 | 48.8 | 63.8 | 65.2 | 48.8 | 50.4 | 74.8 | 77.6 |
| must | Given "[p]" must you reason that "[h]"? | 58.4 | 52.8 | 51.4 | 54.8 | 65.2 | 63.2 | 54.2 | 53.2 | 69.8 | 67.6 |
| must | Must the relationship between "[p]" and "[h]" be ent.? | 48.0 | 41.2 | 53.4 | 56.0 | 62.4 | 62.8 | 49.2 | 48.4 | 67.6 | 66.0 |
| must | Given "[p]" must it be true that "[h]"? | 49.4 | 50.0 | 51.4 | 55.2 | 66.8 | 62.0 | 56.8 | 51.6 | 74.4 | 69.6 |
| must | "[p]" Using only the above description must "[h]" be correct? | 52.4 | 48.0 | 53.8 | 54.8 | 63.6 | 58.0 | 55.0 | 76.4 | 68.8 | 74.4 |
| should | Given "[p]" should you assume that "[h]"? | 50.8 | 50.8 | 52.2 | 59.2 | 63.6 | 62.8 | 51.6 | 59.6 | 70.6 | 66.8 |
| should | Given "[p]" should the claim "[h]" be correct? | 53.4 | 48.0 | 50.2 | 52.4 | 65.0 | 58.8 | 53.8 | 68.8 | 70.2 | 78.0 |
| should | Given "[p]" should you deduce that "[h]"? | 51.2 | 51.6 | 52.4 | 57.2 | 64.4 | 64.4 | 54.0 | 62.8 | 69.8 | 68.0 |
| should | Given "[p]" should it follow that "[h]"? | 54.6 | 53.2 | 52.0 | 56.8 | 64.2 | 60.0 | 56.4 | 61.2 | 70.2 | 67.6 |
| should | Given "[p]" should this imply that "[h]"? | 54.0 | 50.8 | 49.8 | 56.0 | 65.8 | 62.8 | 52.2 | 56.8 | 69.6 | 66.8 |
| should | Given "[p]" should you infer that "[h]"? | 51.6 | 52.8 | 51.6 | 57.2 | 63.8 | 63.6 | 55.8 | 62.8 | 69.6 | 67.6 |
| should | Given "[p]" should you be justified in saying that "[h]"? | 53.6 | 48.4 | 52.6 | 55.6 | 65.4 | 57.2 | 54.8 | 61.6 | 73.8 | 72.8 |
| should | Given premise "[p]" and hypothesis "[h]" should the label be ent.? | 54.2 | 52.0 | 49.4 | 51.2 | 62.8 | 63.6 | 49.2 | 50.4 | 74.6 | 80.4 |
| should | Given "[p]" should you reason that "[h]"? | 52.2 | 48.0 | 52.2 | 57.2 | 65.0 | 62.8 | 55.0 | 60.8 | 71.0 | 70.8 |
| should | Should the relationship between "[p]" and "[h]" be ent.? | 50.0 | 40.8 | 53.2 | 60.4 | 62.6 | 57.6 | 49.2 | 48.8 | 66.8 | 69.2 |
| should | Given "[p]" should it be true that "[h]"? | 55.8 | 52.4 | 51.8 | 55.6 | 66.4 | 59.6 | 53.6 | 61.2 | 71.6 | 73.6 |
| should | "[p]" Using only the above description should "[h]" be correct? | 52.2 | 46.0 | 53.0 | 52.4 | 65.2 | 54.4 | 54.4 | 74.0 | 69.4 | 74.0 |
| would | Given "[p]" would you assume that "[h]"? | 54.8 | 52.4 | 52.2 | 55.6 | 65.6 | 62.8 | 51.4 | 58.4 | 71.8 | 74.0 |
| would | Given "[p]" would the claim "[h]" be correct? | 62.0 | 52.4 | 50.2 | 59.2 | 66.4 | 56.0 | 54.4 | 71.2 | 72.4 | 79.2 |
| would | Given "[p]" would you deduce that "[h]"? | 52.2 | 50.8 | 52.4 | 57.2 | 65.8 | 62.4 | 52.4 | 65.2 | 71.2 | 74.4 |
| would | Given "[p]" would it follow that "[h]"? | 56.2 | 52.8 | 51.6 | 53.6 | 64.8 | 62.8 | 53.6 | 60.0 | 70.0 | 70.4 |
| would | Given "[p]" would this imply that "[h]"? | 55.2 | 58.4 | 50.8 | 57.6 | 65.4 | 60.8 | 51.8 | 54.8 | 70.8 | 70.0 |
| would | Given "[p]" would you infer that "[h]"? | 52.6 | 58.8 | 51.0 | 58.8 | 66.8 | 60.4 | 50.8 | 63.6 | 71.0 | 74.0 |
| would | Given "[p]" would you be justified in saying that "[h]"? | 56.0 | 50.0 | 52.8 | 56.8 | 66.8 | 57.2 | 51.8 | 56.8 | 73.8 | 78.4 |
| would | Given premise "[p]" and hypothesis "[h]" would the label be ent.? | 53.0 | 49.2 | 49.8 | 48.4 | 63.2 | 64.8 | 49.0 | 46.0 | 75.4 | 80.4 |
| would | Given "[p]" would you reason that "[h]"? | 54.6 | 51.6 | 52.0 | 55.6 | 67.2 | 64.0 | 52.6 | 61.2 | 71.2 | 73.6 |
| would | Would the relationship between "[p]" and "[h]" be ent.? | 51.0 | 42.8 | 51.2 | 58.8 | 63.6 | 54.0 | 49.2 | 47.6 | 67.4 | 68.4 |
| would | Given "[p]" would it be true that "[h]"? | 60.8 | 53.2 | 52.2 | 56.4 | 66.8 | 60.4 | 55.2 | 59.6 | 75.2 | 78.4 |
| would | "[p]" Using only the above description would "[h]" be correct? | 56.0 | 47.2 | 51.6 | 54.4 | 64.6 | 59.6 | 54.8 | 77.6 | 70.6 | 77.2 |
| assertion | Given "[p]" is the assertion "[h]" correct? | 54.6 | 52.8 | 52.0 | 58.8 | 66.8 | 57.6 | 53.6 | 69.6 | 73.6 | 78.0 |
| assertion | Given "[p]" is the assertion "[h]" true? | 53.6 | 51.2 | 53.4 | 58.4 | 64.2 | 59.6 | 53.8 | 66.0 | 74.2 | 78.4 |
| claim | Given "[p]" is the claim "[h]" correct? | 53.2 | 48.4 | 52.8 | 59.6 | 67.4 | 59.6 | 52.2 | 66.0 | 73.4 | 79.6 |
| claim | Given "[p]" is the claim "[h]" true? | 55.0 | 50.8 | 53.2 | 57.6 | 66.2 | 61.6 | 52.6 | 66.0 | 73.6 | 78.0 |
| ent. | Given premise: "[p]" and hypothesis: "[h]" is the label ent.? | 54.0 | 49.6 | 50.0 | 47.2 | 62.8 | 63.2 | 49.0 | 46.0 | 74.2 | 83.2 |
| ent. | Is the relationship between "[p]" and "[h]" ent.? | 49.2 | 35.2 | 48.4 | 63.6 | 63.4 | 57.6 | 49.0 | 46.0 | 67.8 | 67.2 |
| implication | Given premise: "[p]" and hypothesis: "[h]" is the label implication? | 55.8 | 60.4 | 52.2 | 56.0 | 63.8 | 65.6 | 49.6 | 46.0 | 73.2 | 84.0 |
| implication | Is the relationship between "[p]" and "[h]" implication? | 53.8 | 48.4 | 49.4 | 64.4 | 63.6 | 62.0 | 48.4 | 46.8 | 66.8 | 66.0 |

Table 14: Detailed list of prompts for Natural Language Inference with accuracy per prompt on RTE and CB across all models. Part 2 of 2.

| property | LLaMA 30b | OPT 1.3b | OPT-IML 1.3b | OPT 30b | OPT-IML 30b |
|---|---|---|---|---|---|
| imperative | **12.94** | **8.64** | **5.19** | **5.39** | **7.75** |
| indicative | 13.01 | 8.68 | 5.33 | 5.42 | 7.76 |
| interrogative | 12.99 | 8.74 | 5.38 | 5.41 | 7.77 |
| active | 13.01 | 8.77 | **5.39** | 5.4 | 7.78 |
| passive | **13.0** | **8.66** | 5.42 | **5.4** | **7.77** |
| past | 12.99 | **8.68** | 5.33 | 5.43 | 7.77 |
| present | 12.99 | 8.71 | 5.35 | **5.39** | 7.77 |
| future | **12.98** | 8.83 | **5.33** | 5.43 | **7.77** |
| can | 12.99 | 8.59 | 5.31 | 5.42 | 7.78 |
| could | 12.99 | 8.6 | 5.34 | 5.4 | 7.78 |
| may | 12.99 | 8.51 | **5.31** | 5.39 | 7.77 |
| might | **12.97** | 8.77 | 5.36 | **5.37** | 7.76 |
| must | 12.99 | **8.48** | 5.32 | 5.39 | **7.76** |
| should | 12.99 | 8.7 | 5.37 | 5.39 | 7.78 |
| would | 12.98 | 8.66 | 5.35 | 5.4 | 7.77 |
| appraisal | **12.95** | 8.74 | 5.38 | 5.42 | **7.76** |
| commentary | 12.97 | 8.73 | 5.36 | 5.4 | 7.76 |
| critique | 12.99 | 8.73 | 5.36 | 5.41 | 7.77 |
| evaluation | 12.99 | 8.74 | 5.38 | 5.41 | 7.77 |
| review | 12.99 | **8.72** | **5.35** | **5.4** | 7.77 |

Table 15: Perplexity scores for prompts on IMDB

| property | LLaMA 30b | OPT 1.3b | OPT-IML 1.3b | OPT 30b | OPT-IML 30b |
|---|---|---|---|---|---|
| imperative | **13.21** | **8.7** | **7.8** | **7.32** | **8.34** |
| indicative | 13.66 | 8.99 | 7.91 | 7.49 | 8.52 |
| interrogative | 13.53 | 8.92 | 8.13 | 7.44 | 8.58 |
| active | 13.68 | 8.99 | 8.17 | **7.5** | 8.65 |
| passive | **13.57** | **8.95** | **8.13** | 7.5 | **8.54** |
| past | 13.57 | 8.87 | 8.03 | 7.43 | 8.56 |
| present | 13.55 | 8.88 | 8.05 | **7.38** | 8.57 |
| future | **13.5** | **8.84** | **7.99** | 7.4 | **8.52** |
| can | 13.53 | 8.91 | 8.03 | 7.42 | 8.54 |
| could | 13.54 | 8.9 | 8.03 | 7.39 | 8.59 |
| may | 13.58 | 8.91 | 8.06 | 7.41 | 8.56 |
| might | **13.46** | 8.87 | **7.98** | **7.34** | 8.54 |
| must | 13.54 | 8.88 | 8.07 | 7.4 | **8.52** |
| should | 13.54 | 8.87 | 8.12 | 7.36 | 8.58 |
| would | 13.46 | **8.86** | 8.05 | 7.39 | 8.56 |
| appraisal | **13.29** | 8.95 | 8.12 | 7.49 | **8.51** |
| commentary | 13.43 | 8.91 | **8.01** | 7.45 | 8.54 |
| critique | 13.53 | 8.91 | 8.09 | 7.48 | 8.57 |
| evaluation | 13.53 | 8.92 | 8.13 | 7.44 | 8.58 |
| review | 13.5 | **8.85** | 8.05 | **7.37** | 8.58 |

Table 16: Perplexity scores for prompts on SST

| property | LLaMA 30b | OPT 1.3b | OPT-IML 1.3b | OPT 30b | OPT-IML 30b |
|---|---|---|---|---|---|
| imperative | **16.25** | **9.88** | **7.96** | 6.31 | 9.36 |
| indicative | 16.4 | 9.98 | 8.05 | 6.35 | 9.43 |
| interrogative | 16.41 | 9.96 | 8.08 | 6.31 | 9.43 |
| active | 15.92 | 9.74 | 7.99 | 6.42 | 9.21 |
| passive | **15.82** | **9.7** | **7.93** | 6.42 | **9.19** |
| past | 16.4 | 10.03 | 8.14 | 6.33 | 9.44 |
| present | 16.4 | 10.02 | 8.13 | **6.31** | **9.43** |
| future | **16.38** | **10.02** | **8.12** | 6.31 | 9.43 |
| can | 16.4 | 9.96 | **8.06** | 6.31 | 9.42 |
| could | 16.39 | **9.93** | 8.07 | 6.31 | 9.42 |
| may | 16.42 | 9.94 | 8.07 | 6.33 | 9.44 |
| might | 16.39 | 9.95 | 8.07 | 6.31 | 9.43 |
| must | 16.41 | 9.96 | 8.07 | 6.32 | 9.43 |
| should | 16.4 | 9.95 | 8.09 | **6.3** | 9.44 |
| would | **16.38** | 9.94 | 8.08 | 6.32 | **9.41** |
| assertion | 16.47 | **10.05** | **8.19** | 6.31 | **9.43** |
| claim | **16.45** | 10.06 | 8.2 | **6.3** | 9.43 |
| implication | 16.29 | 10.0 | 8.04 | 6.39 | 9.46 |
| entailment | **16.25** | **9.97** | **8.04** | **6.38** | **9.43** |

Table 17: Perplexity scores for prompts on CB

| property | LLaMA 30b | OPT 1.3b | OPT-IML 1.3b | OPT 30b | OPT-IML 30b |
|---|---|---|---|---|---|
| imperative | **15.53** | **9.88** | **8.48** | 6.34 | **9.08** |
| indicative | 15.7 | 10.01 | 8.55 | 6.38 | 9.13 |
| interrogative | 15.68 | 9.95 | 8.57 | **6.3** | 9.13 |
| active | 15.33 | 9.74 | 8.49 | 6.43 | 8.99 |
| passive | **15.22** | **9.69** | **8.42** | **6.41** | **8.97** |
| past | 15.62 | 9.84 | 8.49 | 6.28 | 9.12 |
| present | 15.61 | 9.82 | 8.48 | 6.27 | **9.1** |
| future | **15.59** | **9.79** | **8.46** | **6.26** | 9.1 |
| can | 15.59 | 9.87 | **8.52** | 6.26 | 9.1 |
| could | 15.58 | 9.87 | 8.53 | 6.26 | 9.1 |
| may | 15.61 | **9.83** | 8.53 | 6.27 | 9.12 |
| might | 15.58 | 9.87 | 8.53 | 6.26 | 9.11 |
| must | 15.6 | 9.88 | 8.53 | 6.27 | 9.11 |
| should | 15.59 | 9.88 | 8.56 | **6.25** | 9.12 |
| would | **15.57** | 9.88 | 8.54 | 6.27 | **9.09** |
| assertion | 15.89 | **9.84** | **8.54** | 6.26 | **9.12** |
| claim | **15.88** | 9.85 | 8.54 | **6.24** | 9.12 |
| entailment | **15.6** | **9.73** | 8.35 | **6.33** | **9.09** |
| implication | 15.64 | 9.75 | **8.35** | 6.34 | 9.12 |

Table 18: Perplexity scores for prompts on RTE

| property | LLaMA 30b | OPT 1.3b | OPT-IML 1.3b | OPT 30b | OPT-IML 30b |
|---|---|---|---|---|---|
| imperative | **13.47** | 8.67 | 7.34 | 5.93 | **8.39** |
| indicative | 13.55 | 8.64 | 7.32 | 5.94 | 8.41 |
| interrogative | 13.5 | **8.63** | **7.3** | **5.92** | 8.4 |
| active | **13.48** | **8.61** | **7.28** | **5.87** | **8.37** |
| passive | 13.53 | 8.64 | 7.3 | 5.94 | 8.39 |
| past | 13.48 | 8.58 | 7.29 | 5.9 | **8.35** |
| present | 13.46 | 8.61 | 7.29 | **5.88** | 8.38 |
| future | **13.46** | **8.58** | **7.27** | 5.88 | 8.41 |
| can | 13.43 | 8.62 | 7.27 | 5.87 | **8.36** |
| could | 13.44 | 8.61 | 7.28 | 5.88 | 8.39 |
| may | 13.43 | 8.61 | 7.29 | 5.88 | 8.41 |
| might | 13.44 | 8.61 | 7.28 | 5.88 | 8.41 |
| must | 13.45 | 8.63 | **7.25** | 5.88 | 8.37 |
| should | 13.43 | 8.6 | 7.28 | **5.86** | 8.43 |
| would | **13.41** | **8.6** | 7.25 | 5.86 | 8.4 |
| appropriate | 13.49 | 8.62 | 7.3 | 5.88 | 8.41 |
| correct | **13.47** | **8.6** | 7.27 | **5.86** | **8.37** |
| proper | 13.5 | 8.63 | 7.29 | 5.88 | 8.41 |
| right | 13.47 | 8.61 | **7.25** | 5.87 | 8.37 |
| answer | **13.47** | **8.61** | **7.27** | **5.86** | **8.37** |
| reply | 13.5 | 8.66 | 7.33 | 5.91 | 8.43 |
| response | 13.49 | 8.63 | 7.31 | 5.9 | 8.42 |
| solution | 13.5 | 8.63 | 7.3 | 5.92 | 8.4 |

Table 19: Perplexity scores for prompts on ARC-E

| property | LLaMA 30b | OPT 1.3b | OPT-IML 1.3b | OPT 30b | OPT-IML 30b |
|---|---|---|---|---|---|
| imperative | 14.16 | 8.84 | 6.77 | 5.66 | 8.29 |
| indicative | 14.19 | 8.84 | 6.81 | 5.61 | 8.33 |
| interrogative | **14.14** | **8.82** | **6.74** | **5.6** | **8.26** |
| active | **14.14** | **8.82** | 6.75 | **5.6** | **8.27** |
| passive | 14.17 | 8.84 | 6.78 | 5.64 | 8.28 |
| past | 14.15 | 8.89 | 6.78 | 5.6 | 8.28 |
| present | 14.15 | **8.82** | 6.75 | **5.59** | **8.28** |
| future | **14.15** | 8.89 | 6.78 | 5.6 | 8.28 |
| can | 14.15 | 8.84 | 6.76 | 5.59 | 8.27 |
| could | 14.15 | 8.82 | 6.76 | 5.61 | **8.26** |
| may | 14.15 | 8.82 | 6.76 | 5.6 | 8.28 |
| might | 14.15 | 8.82 | 6.77 | 5.6 | 8.27 |
| must | 14.16 | 8.83 | 6.76 | 5.6 | 8.29 |
| should | 14.15 | 8.82 | 6.75 | **5.58** | 8.28 |
| would | **14.14** | **8.81** | **6.75** | 5.59 | 8.26 |
| appropriate | 14.15 | 8.83 | 6.75 | 5.61 | 8.28 |
| correct | **14.14** | **8.81** | 6.75 | **5.59** | 8.26 |
| proper | 14.16 | 8.83 | 6.75 | 5.61 | 8.28 |
| right | 14.14 | 8.82 | **6.74** | 5.6 | **8.26** |
| answer | **14.14** | **8.82** | **6.75** | **5.6** | **8.27** |
| reply | 14.16 | 8.83 | 6.79 | 5.62 | 8.29 |
| response | 14.16 | 8.83 | 6.77 | 5.61 | 8.28 |
| solution | 14.16 | 8.84 | 6.77 | 5.62 | 8.29 |

Table 20: Perplexity scores for prompts on BoolQ