# OpenReview forum: "The language of prompting: What linguistic properties make a prompt successful?"
_EMNLP/2023/Conference — EMNLP 2023 Findings_

### Official Review · Reviewer_z3Ht · 2023-08-03

**Soundness:** 3

**Excitement:**

2: Mediocre: This paper makes marginal contributions (vs non-contemporaneous work), so I would rather not see it in the conference.

**Missing References:**

No missing references

**Paper Topic And Main Contributions:**

This is an experimental paper that aims to investigate the following research question:
Do linguistic variations in prompts have an effect in performance over final NLP tasks for LLMs?
To pursue its goal, the paper examined the performance of LLMs on different classification tasks mainly contained in GLUE and SuperGLUE. Linguistic variations are characterized by different features: mood, aspect, tense, modality, and synonymy.

**Questions For The Authors:**

- Can you please elaborate on the reason why you have decided to not apply synonymy to the main verb of the prompt?


**Reasons To Accept:**

- The paper aims to make a significant contribution to the understanding of how prompts should be phrased in order to obtain the best results over linguistic tasks.

- The paper performs extensive experimental analysis

**Reasons To Reject:**

- The large experimental analysis does not reveal any clear pattern and does not help in choosing what are the best ways to write a prompt for a classification task

- The analysis is confined to classification tasks

- It is even questionable if it is important to perform a linguistic classification task with LLMs. Indeed, solving many of the tasks is useless since we have LLMs that make decisions and generate responses without using explicit solvers for these tasks.

**Reproducibility:**

4: Could mostly reproduce the results, but there may be some variation because of sample variance or minor variations in their interpretation of the protocol or method.

**Reviewer Confidence:**

4: Quite sure. I tried to check the important points carefully. It's unlikely, though conceivable, that I missed something that should affect my ratings.

**Typos Grammar Style And Presentation Improvements:**

Writing the introduction:

- please reorganize the introduction such that there is a clear paragraph describing the work of your paper, which is: studying the effect on performances of syntactic variations on semantically equivalent prompts.
- "To test ***this hypothesis***, we examine prompting ... " (line 056). It is weird to start a paragraph with an anaphoric word referring to an idea expressed in the previous paragraph. This sentence needs rephrasing.
- Make clear that the "hypothesis" is expressed in lines 051-055 (that is **this hypothesis***) and it is part of your intuition in building the paper

---

> ### Author Rebuttal · Authors · 2023-08-28
>
> Thank you for the review, and feedback on our paper! We’d like to offer the following clarifications on your inputs and questions:
>
> *Re presentation improvements:* Thank you for your feedback on the phrasing of the introduction in lines 0-56, we will adapt this in an updated version of the paper.
>
> *Re Q on synonymy for main verb of prompt:* We experimented with synonyms for the main verb of the instruction, but did not include the results, since usually there were more synonyms available for other content words in the sentence, so these results were more interesting. We will include the results in the camera-ready version of the paper.
>
> *Re comment 1:* The main focus of the paper was not to aid in choosing the best prompt for a task, but systematically examine current evaluation practices by analysing model robustness to linguistic variation. We thus take a novel perspective on the question of model robustness to prompt variability, and based on our results provide a set of recommendations for a more rigorous evaluation of model prompting. We believe that these findings and recommendations would help the field by critically examining commonly held assumptions and instigating a conversation on what transparent results reporting looks like for prompt-based evaluation.
>
> *Re comments 2, 3:* We acknowledge that LLMs can in principle solve tasks in an open-ended way, i.e., we can prompt them with an instruction and have them generate full sentences that will contain an answer. But such an open-ended generative set-up is very difficult to evaluate efficiently. That is why we chose to predict based on the maximum log probability among answer options. Is this what you meant by ‘explicit solvers’? With our evaluation method we are following what is the standard prompt-based evaluation procedure for LLMs (see Sanh et al. 2022, Wei et al. 2022, Webson and Pavlick 2022, i.a.). Due to surface form competition (Holtzman et al. 2021) this is the evaluation method of choice also for many QA tasks.
> In our work, we aimed to have a diverse set of NLU tasks with a consistent evaluation procedure across tasks, and hence did not include open-ended generation tasks. We agree however that adding additional tasks, such as QA, would strengthen the paper. We are working on including those additional results in an updated version of the paper.
>
> ------------------------------------------
> References:
> * Sanh, Victor, et al. "Multitask Prompted Training Enables Zero-Shot Task Generalization." ICLR 2022-Tenth International Conference on Learning Representations. 2022.
>
> * Wei, Jason, et al. "Finetuned Language Models are Zero-Shot Learners." (2022).
>
> * Webson, Albert, and Ellie Pavlick. "Do Prompt-Based Models Really Understand the Meaning of Their Prompts?." Proceedings of the 2022 Conference of the North American Chapter of the Association for Computational Linguistics: Human Language Technologies. 2022.
>
> * Holtzman, Ari, et al. "Surface Form Competition: Why the Highest Probability Answer Isn’t Always Right." Proceedings of the 2021 Conference on Empirical Methods in Natural Language Processing. 2021.

---

### Official Review · Reviewer_Nfib · 2023-08-03

**Soundness:** 4

**Excitement:**

2: Mediocre: This paper makes marginal contributions (vs non-contemporaneous work), so I would rather not see it in the conference.

**Paper Topic And Main Contributions:**

This paper conducts comprehensive experiments to show the prompt instability of LLMs. Unlike previous work, this paper investigates the prompt robustness of the model from a lexical perspective, where the authors slightly shift the linguistic properties (e.g., mood, tense) of prompt and find a significant performance variation of LLMs, including instruction-tuned LLMs. Some results are also different from the previous works, suggesting the vulnerability of numerous research findings in current prompt-related research. Thus, the author also concludes with some empirical advice for future research.

**Questions For The Authors:**

In your experiments, did you design all the prompts by yourself? Since you want to investigate the prompt robustness of instruction-tuned LMs, it should be better to choose the prompts used in tuning OPT-IML (seen by LMs during training), which can more effectively demonstrate your motivations.

**Reasons To Accept:**

1. **A novel perspective**. The author utilizes a lexico-level perturbation and linguistic perspective to investigate the prompt robustness, which differs from the previous sentence-level paraphrasing or simple synonymy replacing.
2. **Useful proposal**. The proposal concluded by this paper might also benefit future research.

**Reasons To Reject:**

1. **Limited contribution**. Though this paper provides some useful suggestions, most conclusions have already been proposed by a bunch of previous works or are just commonsense for the community (e.g., prompt transfer). During the whole section 5, the authors simply introduce the results and compare scores, but there are no more insights, analysis, or explanations. After reading the whole paper, I didn't seem to learn much from it.

2. **Concern about the experiment setting.** In the experiment, the authors adopt five models, but essentially, there are only two model categories --- vanilla LMs and instruction-tuned LMs. Similarly, all of these models (OPT, LLaMA) are decoder-only LMs. I am worried about the expandability of the experimental conclusion. Similarly, the datasets seem only cover the cross-dataset generalization setting; a better choice is to adopt a more challenging cross-task setting [1], which is a popular trend in the current community.

3. **Some viewpoints are subjective.** For example, in line 537, the authors suggest "include estimates of performance mean and variance based on a large set of prompts". Personally, I think the prompt is just a fundamental "feature" for LLMs to deal with a specific task, "prompt engineering" is just similar to "feature engineering"; we have to tune an optimal (or sub-optimal) prompt for one specific task, but we cannot anticipate the LMs will be robust to the change of the prompt. So reporting statistic scores on large-set prompts for one task seems unrealistic. But I agree with "treating prompts as hyperparameters" --- tune the prompt on dev set, fix it, and then repot scores on the test set.

4. **Writing.** I found it hard to follow some specific sections of this paper. For example, the title of section 5.3 --- "Robustness and instruction-tuning" --- sounds a little bit confusing.

---

References:

[1]. Mishra S, Khashabi D, Baral C, et al. Cross-task generalization via natural language crowdsourcing instructions[J]. arXiv preprint arXiv:2104.08773, 2021.

**Reproducibility:**

4: Could mostly reproduce the results, but there may be some variation because of sample variance or minor variations in their interpretation of the protocol or method.

**Reviewer Confidence:**

4: Quite sure. I tried to check the important points carefully. It's unlikely, though conceivable, that I missed something that should affect my ratings.

---

> ### Author Rebuttal · Authors · 2023-08-28
>
> Thank you for the review, for appreciating the novel perspective of our work and giving some actionable suggestions! We’d like to offer the following clarifications regarding your feedback and questions:
>
> *Re comment 1:* We agree that the fact that prompts transfer poorly is generally known. We deemed it valuable to investigate this thoroughly, systematically and comprehensively, since it is not known if there’s an upper bound to performance variation. In particular, instruction-tuning is often seen as a fix to this without rigorous justification. That is why we prompted, in particular, instruction tuned models with a large set of prompts to demonstrate how much performance can vary, even on seen datasets.
>
> We agree that section 5 would benefit from an additional discussion and analysis. If the paper is accepted, we will use the additional page in the camera-ready version to expand the discussion of the relationships between different linguistic properties in the prompts and their effect on performance, e.g. by using hierarchical models which are commonly applied in the behavioural sciences and were recently also employed for the analysis of LMs (Lampinen et al. 2022).
>
> *Re comment 2:* Thank you for your input on the experimental set-up. We initially chose instruction-tuned and non-instruction-tuned decoder-only models as the scope of our paper, since decoder-only models are most widely used at the moment. We re-ran some of our experiments using an encoder-decoder model, Flan-T5 (XL), and found that average performance per linguistic property still varied between properties, similarly to what we reported on the encoder-only models in the paper.
>
> | Mood  | sst | imdb | rte | hans |
> |----------------|----|------|--------|-------------|
> Interrogative | 87.3 | 95.8 | 88.8 | 77.5 |
> Indicative      | 86.6 | 94.6 | 86.1 | 76.4 |
> Imperative     | 88.2 | 96.1 | 90.9 | 77.3 |
>
>
> |Synonymy | rte | hans |
> |----|----------|---------------|
> Entailment | 95 | 77 |
> Implication | 96 | 78.4 |
> Assertion | 91 | 81.5 |
> Claim | 92.5 | 81 |
>
> |Synonymy | sst | imdb |
> |---|---|---|
> Appraisal | 83.4 | 85.7 |
> Commentary | 77.7 | 80.4 |
> Critique | 81.5 | 85.8 |
> Evaluation | 83.7 | 88.4 |
>
> If the paper is accepted, we will include results on Flan-T5 in the camera ready version of the paper. We would also very much welcome your advice on which other types of models you would recommend including.
> Moving to the cross-task setting would indeed be a valuable avenue for future work. For this paper, we chose a cross-dataset setup, since the focus was comparing the same prompts on multiple dataset of the same task.
>
> *Re comment 3:*  The focus of our recommendations is on how to conduct evaluation in a more rigorous fashion, and our position is that a metric quantifying the dependence of model performance on a particular prompt formulation is a useful way to make progress in understanding of how to perform prompt engineering, be it manual engineering or automatic prompt optimization.
>
> *Re comment 4:* Thank you for the comment, we will rephrase the heading of section 5.3. for clarity. If this is possible, would you be able to offer more advice on which sections you found hard to follow? This would help us a lot in improving the paper.
>
> *Re question on prompt design:* Yes, we did construct the prompts ourselves in order to minimise the potential sources of variation and have a more homogeneous set of prompts in terms of sentence structure. We found prompts in public instruction tuning sets to vary quite widely. We do agree that experimenting with the exact prompts from OPT-IML would strengthen the paper and will include those experiments in an updated version of the paper. This is good advice, thank you!
>
> ------------------------------------------------------
> References:
>
> * Lampinen, Andrew, et al. "Can language models learn from explanations in context?." Findings of the Association for Computational Linguistics: EMNLP 2022. 2022.

---

### Official Review · Reviewer_4Bot · 2023-08-04

**Soundness:** 4

**Excitement:**

4: Strong: This paper deepens the understanding of some phenomenon or lowers the barriers to an existing research direction.

**Missing References:**

* Do Prompts Solve NLP Tasks Using Natural Language? Sen Yang, Yunchen Zhang, Leyang Cui and Yue Zhang. ArXiv 2022.

**Paper Topic And Main Contributions:**

This paper presents a large-scale systematic analysis of how the variations in prompts influence downstream performances. The authors conduct empirical studies on multiple tasks using different LMs under various settings. In these studies, the authors control (i) grammatical properties such as mood, tense and modality; and (ii)  lexico-semantic variation by replacing a word with its synonyms. Experimental results show that prompts transfer rather poorly across datasets and LMs.

**Reasons To Accept:**

* A very comprehensive and systematic empirical study of the instability of prompt choices. Those experimental results could potentially be utilized in many future studies.
* The authors present several interesting conclusions, such as the existence of the correlation between LM perplexity of prompts and performance.

**Reasons To Reject:**

* Though it is not a serious flaw, it would make this paper stronger if the authors could give some explanations on why they choose certain types of linguistic variations, such as mood and tense. What if the prompts are paraphrased to a larger extent, e.g., using a constituency parser to generate sub-tree structures and making changes on the sub-tree level?

**Reproducibility:**

4: Could mostly reproduce the results, but there may be some variation because of sample variance or minor variations in their interpretation of the protocol or method.

**Reviewer Confidence:**

4: Quite sure. I tried to check the important points carefully. It's unlikely, though conceivable, that I missed something that should affect my ratings.

---

> ### Author Rebuttal · Authors · 2023-08-28
>
> Thank you for your time, your helpful suggestions, and appreciation of our paper!
>
> Thank you for pointing us to the missing reference, we will include it in an updated version of the paper.
>
> Regarding your question, we didn't paraphrase instructions to a larger extent, because we wanted to vary one linguistic property at a time and keep the variation between sentences minimal, in order to not introduce different sources of variation at once. Hence, we constructed all our prompts to only consist of a main clause with varying tense, mood, etc. Thank you for bringing this up, we will clarify this in an updated version of the paper!

---

### Meta-Review · Area_Chair_8kCV · 2023-09-25

**Recommendation:** 3

**Metareview:**

This paper examines how linguistic variations in prompts affect the performance of large language models (LLMs) on various classification tasks. The paper tests different LLMs, including instruction-tuned LLMs, on datasets from GLUE and SuperGLUE. The paper manipulates the prompts by changing their grammatical properties (such as mood, tense, and modality) and lexical properties (such as synonyms). The paper finds that the prompts are unstable and sensitive to these variations, and that the performance varies significantly across datasets and LLMs. The paper provides some empirical suggestions for future prompt-related research. The paper elaborates upon the prompt variations aptly, is well written and definitely has merits for downstream use-cases, the scope limited to classification based tasks only for a long paper.

---

### Decision · Program_Chairs · 2023-10-07

**Decision:**

Accept-Findings

**Comment:**

This paper examines how linguistic variations in prompts affect the performance of large language models (LLMs) on various classification tasks. The paper tests different LLMs, including instruction-tuned LLMs, on datasets from GLUE and SuperGLUE. The paper manipulates the prompts by changing their grammatical properties (such as mood, tense, and modality) and lexical properties (such as synonyms). The paper finds that the prompts are unstable and sensitive to these variations, and that the performance varies significantly across datasets and LLMs. The paper provides some empirical suggestions for future prompt-related research. The paper elaborates upon the prompt variations aptly, is well written and definitely has merits for downstream use-cases, the scope limited to classification based tasks only for a long paper.